



# What millimeter-wavelength radar reflectivity reveals about snowfall: An information-centric analysis

Norman B. Wood[1] and Tristan S. L'Ecuyer[2]

[1]Space Science and Engineering Center, University of Wisconsin - Madison, Madison, WI
[2]Department of Atmospheric and Oceanic Sciences, University of Wisconsin - Madison, Madison, WI

**Correspondence:** Norman B. Wood (norman.wood@ssec.wisc.edu)

**Abstract.** The ability of single-frequency, millimeter-wavelength radar reflectivity observations to provide useful constraints for retrieval of snow particle size distribution (PSD) parameters, snowfall rates, and snowfall accumulations is examined. An optimal estimation snowfall retrieval that allows analyses of retrieval uncertainties and information content is applied to observations of near-surface W-band reflectivities from multiple snowfall events during the 2006-2007 winter season in southern

Ontario. Retrieved instantaneous snowfall rates generally have uncertainties greater than 100%, but single-event and seasonal snow accumulations from the retrieval results match well with independent collocated measurements of accumulations. Absolute fractional differences are mainly below 30% for individual events that have more substantial accumulations and, for the season, 12.6%. Uncertainties in retrieved snowfall rates are driven mainly by uncertainties in the retrieved PSD parameters, followed by uncertainties in particle model parameters and, to a lesser extent, the uncertainties in the fallspeed model. Un-

certainties attributable to assuming an exponential distribution are negligible. The results indicate that improvements to PSD and particle model a priori constraints provide the most impactful path forward for reducing uncertainties in retrieved snowfall rates. Information content analyses reveal that PSD slope is well-constrained by the retrieval. Given the sensitivity of PSD slope to microphysical transformations, the results show that such retrievals, when applied to radar reflectivity profiles, could provide information about microphysical transformations in the snowing column. The PSD intercept is less well constrained

by the retrieval. While applied to near-surface radar observations in this study, the retrieval is applicable as well to radar observations aloft, such as those provided by profiling ground-based, airborne, and satellite-borne radars under lighter snowfall conditions when attenuation and multiple scattering can be neglected.

## 1 Introduction

Radar observations focused on snowfall from platforms outside the established weather surveillance radar networks have become ubiquitous over the last two decades, largely due to increased interest in the role of snowfall in mid- and high-latitude microphysics, hydrology and climate. This research accelerated with the advent of satellite-borne radars flown by missions



to quantify global hydrometeor and precipitation properties. These satellite-borne radars (specifically the CloudSat mission's Cloud Profiling Radar (CPR) (Tanelli et al., 2008) and the Global Precipitation Measurement (GPM) mission's Dual-frequency

Precipitation Radar (DPR) (Toyoshima et al., 2015), with two others anticipated to launch in the coming decade) are capable solely of measuring vertical profiles of radar reflectivity factor (hereafter, reflectivity) along with path integrated attenuation under certain conditions. To understand the capabilities of these satellite-borne radars for quantifying snowfall, we must know how well radar reflectivity observations constrain snowfall properties.

To these ends, CloudSat and GPM have contributed to multiple field experiments involving ground-based radars and de-
signed to provide, in part, ground validation data for the radar remote sensing of snowfall: the Canadian CloudSat-CALIPSO Validation Project (C3VP) (Hudak et al., 2006), the Global Precipitation Measurement (GPM) Cold-season Precipitation Experiment (GCPEx)[(Skofronick-Jackson et al., 2015), the Light Precipitation Validation Experiment (LPVEx) (Petersen et al., 2011), the International Collaborative Experiment during the PyeongChang 2018 Olympics and Paralympics (ICE-POP) (Chandrasekar et al., 2019), and the Olympic Mountains Experiment (OLYMPEx) (Houze et al., 2017). Ground-based pro-
filing radars such as the METEK Micro Rain Radar (MRR) (Klugmann et al., 1996), which operates at the transition between millimeter and centimeter wavelengths, have been key components of these larger field experiments. In addition, a number of smaller, more focused field campaigns (Pettersen et al., 2020; Schirle et al., 2019) deployed MRRs exclusively for vertically profiling frozen precipitation. While these ground-based radars may provide additional capabilities such as Doppler velocity measurement, their reflectivity measurements in snowfall are a valuable resource for examining the capabilities of the satellite-
borne radars (Maahn et al., 2014).

The ability of radar reflectivity to constrain snowfall properties, however, has not been well evaluated. Snowfall exhibits a wide range of microphysical characteristics that influence radar reflectivity and snowfall rate. Most notable to casual observers are variations in particle habits: pristine dendrites, needles, columns, plates and bullets; aggregates of the same; pellets and graupel for example. Underlying these differences in habit are variations in mass, and, given a particular mass, variations in
how mass is distributed within the particle. Unlike longer-wavelength radars for which radar backscattering properties of snow particles are sensitive primarily to particle mass, at millimeter wavelengths those properties are additionally sensitive to particle shape. Investigations of particle mass and area (an aspect of shape) (Kajikawa, 1972, 1975, 1982; Zikmunda and Vali, 1972, 1977; Heymsfield, 1972; Locatelli and Hobbs, 1974; Mitchell et al., 1990; Mitchell, 1996; Heymsfield and Miloshevich, 2003) have painstakingly determined the broad extent of these variations. Along with differences in single-particle properties,
populations of falling snow particles vary substantially in their concentrations with size (i.e., the spectral particle size distribution, PSD) based on measurements from the ground (e.g., Nakada and Terada, 1935; Imai et al., 1955; Gunn and Marshall, 1958; Rogers, 1973; Brandes et al., 2007) and more recently, with the advent of imaging particle probes, from aircraft (e.g., Passarelli, 1978; Gordon and Marwitz, 1984, 1986; Braham, 1990; Woods et al., 2008; Heymsfield et al., 2008, and references therein). The observed particle concentrations vary over several orders of magnitude.

In radar-based remote sensing scenarios when these properties are not known, these variations produce uncertainty in the relationship between radar reflectivity factor (hereafter, reflectivity) and associated water content and snowfall rate. A common approach to evaluating this uncertainty has been to evaluate modeled reflectivities, water contents and snowfall rates using





a range of assumed particle models and PSDs. The results are often expressed using relationships between reflectivity and snowfall rate ("Z-S" relationships) (Liu, 2008; Kulie and Bennartz, 2009; Matrosov et al., 2008, 2009). This approach allows
the uncertainty in a retrieved snowfall rate to be estimated, but the existing studies have not provided insight into the dominant sources of uncertainty nor into the ability of observed radar reflectivity to constrain various properties controlling the snowfall rate. Posselt et al. (2015) examined uncertainties and information content for radar observations of mixed- and ice-phase regions of a convective storm, but targeted radar systems with more advanced capabilities. Mascio and Mace (2017) used CloudSat and aircraft observations to assess how uncertainties in the ice particle mass-dimension relationship contribute
to radar reflectivity forward model uncertainties, but used known, observed particle size distributions and did not examine the influence of the mass-dimension uncertainties on snowfall retrieval performance.

In this work, we provide uncertainty and information content analyses for retrieving snowfall from observations of radar reflectivity at millimeter wavelengths, focusing on W-band (94 GHz). The results are representative of the general problem of estimating snowfall from such remote radar reflectivity observations without supplementary collocated observations of snow par-
ticle mass-dimension relationships, fallspeeds, and particle size distributions. The results apply particularly to observations by the CPR (Tanelli et al., 2008) and by the DPR's Ka-band radar, but also to reflectivity measurements from ground-based radars such as the MRR (Klugmann et al., 1996) and the Department of Energy Atmospheric Radiation Measurement (ARM) program's Millimeter Wavelength Cloud Radar (Moran et al., 1998) and Ka-band ARM Zenith Radar (KAZR) (Bharadwaj et al., 2013). Our objectives are to identify the snowfall properties that are best constrained by such observations and the most signif-
icant sources of uncertainty in the radar retrieval of snowfall. The results establish a performance baseline for reflectivity-only observations of snowfall, indicate where uncertainty reduction efforts should be focused, and suggest what improvements to radar observing systems could be most beneficial.

The analyses use the optimal estimation (OE) retrieval technique (Rodgers, 2000), which inherently diagnoses information content and uncertainties in retrieved quantities subject to specified uncertainties in measurements, forward models, and a priori
knowledge of the quantities to be retrieved (L'Ecuyer et al., 2006; Cooper et al., 2006). The retrieval produces best-estimates of snow size distribution parameters by using the radar reflectivity observations to refine a priori estimates of those parameters (Sect. 2). The information content metrics provided by OE require all sources of uncertainties in the retrieval process to be specified. These are discussed in Sect. 3. Ground-based radar and precipitation observations allow the retrieval to be tested, showing that size distribution width is best constrained by the retrieval and that uncertainties in retrieved size distribution parameters
(but not uncertainties due to the assumed exponential form of the PSD itself) are the strongest contributors to uncertainties in estimated snowfall rates (Sect. 4). The results suggest that the retrieved size distribution widths could be useful for diagnosing changes in PSD resulting from microphysical processes (Lo and Passarelli, 1982) and that improved observational constraints on size distribution parameters, as might be provided by dual-wavelength radar observations (Matrosov, 1998), would likely enhance snowfall retrieval performance (Sect. 5).


## 2  Retrieval method

The retrieval uses measurements of reflectivity to estimate snow microphysical properties and to quantify water content and snowfall rate. At the wavelengths characteristic of cloud radars such as CloudSat and shorter-wavelength precipitation radars, scattering by precipitation-sized particles does not follow the Rayleigh approximation, and both attenuation and multiple scattering may affect the radar signal. At these wavelengths, snow particle scattering and extinction properties depend not only on mass, but on shape as well. With even simple parameterized expressions for particle mass, shape, and size distribution (PSD), single-frequency observations of radar reflectivity alone are insufficient to reasonably constrain the resulting set of parameters.

To address this insufficiency, retrievals must incorporate a priori information about particle microphysical and scattering properties. This is accomplished here using OE (Rodgers, 2000), a Bayesian technique that allows a priori information to be included explicitly. The input for this retrieval is the $Z_e$ observed by the radar for a range gate identified as containing snow. For notational consistency with other work, we show this as a vector:

$$\boldsymbol{y} = \left[ Z_{e,1}^{obs} \right]. \tag{1}$$

A forward model $\boldsymbol{F}(\boldsymbol{x},\tilde{\boldsymbol{b}})$ relates $\boldsymbol{y}$ to $\boldsymbol{x}$, a state vector of unknown properties to be retrieved, as

$$\boldsymbol{y} = \boldsymbol{F}(\boldsymbol{x},\tilde{\boldsymbol{b}}) + \boldsymbol{\epsilon}, \tag{2}$$

where $\tilde{\boldsymbol{b}}$ are parameters not being retrieved but which influence the forward model results. The forward model approximates the true physical relation between $\boldsymbol{x}$ and $\boldsymbol{y}$, and there are uncertainties associated with both the observations $\boldsymbol{y}$ and the forward model parameters $\tilde{\boldsymbol{b}}$. $\boldsymbol{\epsilon}$ represents the total uncertainty due to all sources. OE attempts to find $\hat{\boldsymbol{x}}$, an estimate of the state which maximizes the posterior conditional probability density function (PDF) $P(\boldsymbol{x} \mid \boldsymbol{y})$, subject also to prior knowledge about the values of $\boldsymbol{x}$. This prior knowledge is described by expected values $\boldsymbol{x}_a$ and their covariances $\mathbf{S}_a$. Assuming Gaussian statistics for the model-measurement errors and the a priori state, minimizing the cost function

$$\Phi(\boldsymbol{x},\boldsymbol{y},\boldsymbol{x}_a) = \left( \boldsymbol{y} - \boldsymbol{F}(\boldsymbol{x},\tilde{\boldsymbol{b}}) \right)^{\mathsf{T}} \mathbf{S}_\epsilon^{-1} \left( \boldsymbol{y} - \boldsymbol{F}(\boldsymbol{x},\tilde{\boldsymbol{b}}) \right) + (\boldsymbol{x} - \boldsymbol{x}_a)^{\mathsf{T}} \mathbf{S}_a^{-1} (\boldsymbol{x} - \boldsymbol{x}_a), \tag{3}$$

with respect to $\boldsymbol{x}$ gives this PDF, where $\mathbf{S}_\epsilon$ is the covariance matrix representing the uncertainties $\boldsymbol{\epsilon}$. The Gaussian assumption is reasonable if the expected values and covariance matrices are known for the model-measurement uncertainties and the a priori state, but other details are lacking. In that case, the Gaussian form maximizes the entropy of a PDF (Shannon and Weaver, 1949; Rodgers, 2000). Assuming an alternate form would introduce constraints on the retrieval that are not justified based on the limited knowledge of the PDF.

Provided the forward model is not excessively nonlinear, Newtonian iteration

$$\hat{\boldsymbol{x}}_{i+1} = \hat{\boldsymbol{x}}_i + \left( \mathbf{S}_a^{-1} + \mathbf{K}_i^{\mathsf{T}} \mathbf{S}_\epsilon^{-1} \mathbf{K}_i \right)^{-1} \left[ \mathbf{K}_i^{\mathsf{T}} \mathbf{S}_\epsilon^{-1} \left( \boldsymbol{y} - \boldsymbol{F}(\hat{\boldsymbol{x}}_i,\tilde{\boldsymbol{b}}) \right) - \mathbf{S}_a^{-1}(\hat{\boldsymbol{x}}_i - \boldsymbol{x}_a) \right], \tag{4}$$





leads to $\hat{\boldsymbol{x}}$, where $\mathbf{K}$ is the Jacobian of the forward model with respect to $\boldsymbol{x}$, and $\mathbf{K}_i = \mathbf{K}(\hat{\boldsymbol{x}}_i)$. Iteration continues until the
covariance-weighted difference in successive $\hat{\boldsymbol{x}}_i$ is much smaller than the number of state vector elements. At convergence, the
covariance of $\hat{\boldsymbol{x}}$ is obtained as

$$\hat{\mathbf{S}}_x = \left( \hat{\mathbf{K}}^{\mathsf{T}} \mathbf{S}_\epsilon^{-1} \hat{\mathbf{K}} + \mathbf{S}_a^{-1} \right)^{-1},\tag{5}$$

where $\hat{\mathbf{K}} = \mathbf{K}(\hat{\boldsymbol{x}})$. As a diagnostic test of the results, a $\chi^2$ statistic is calculated using the retrieved state vector in Eq. (3). A
value near the number of observations suggests correct convergence. Several metrics, determined from the retrieved state and
based in information theory, provide insight into the retrieval performance; these metrics are presented in Sect. 4.

## 2.1   The forward model

To assess the information provided purely by reflectivity observations, whether from ground-, aircraft-, or space-based radars,
the retrieval ignores attenuation and multiple scattering. This treatment would be appropriate for cases with little intervening
scattering and extinction between the radar and observed snowfall, such as when the radar bin containing the snowfall of
interest is near the radar or under light snowfall conditions. For such a case, the singly-scattered reflectivity $Z_e^{ss}$ as a function
of range $R$ from the radar is given by

$$Z_e^{ss}(R) = \frac{\Lambda^4}{\|K_w\|^2 \pi^5} \int_{D_{min}}^{D_{max}} N(D,R)\sigma_{bk}(D,R)dD\tag{6}$$

where $\sigma_{bk}(D,R)$ is the backscatter cross-section for particle size $D$ at range $R$, $N(D,R)$ is the particle size distribution (PSD)
at range $R$, $\Lambda$ is the radar wavelength, and $K_w$ is the dielectric factor for water.

### 2.1.1   Forward model parameters: snow particle model

Backscattering and extinction cross-sections depend intimately on particle mass, shape and orientation relative to the radar
beam. These properties are highly variable for snow particles, and the approach used here is to specify their PDFs a priori
using best estimates and treat their variability as a source of uncertainty in the retrieval. We adopt the common model (e.g.,
Locatelli and Hobbs, 1974; Mitchell, 1996) in which mass and horizontally-projected area are described using power laws

$$m(D_M) = \alpha D_M^\beta\tag{7}$$

$$A_p(D_M) = \gamma D_M^\sigma\tag{8}$$

on particle maximum dimension, $D_M$, and use the particle properties and shape "B8pr-30" (Wood et al., 2015), an idealized
branched spatial particle that was found to minimize bias in simulated reflectivies versus coincident W-band radar observa-
tions. That work used in-situ and remote sensing observations of snow from C3VP (Hudak et al., 2006) along with previously





reported single-particle measurements to develop best estimates and covariances for the power law parameters $\alpha$, $\beta$, $\gamma$, and $\sigma$. These results then constrained discrete dipole approximation calculations using DDSCAT (Draine and Flatau, 1994) to obtain best estimates of snow particle single-scattering properties and their uncertainties at the desired wavelengths. These a priori descriptions of size-resolved particle mass, $A_p$, $\sigma_{bk}$, $\sigma_{ext}$ and their uncertainties constitute the particle model used in the retrieval

and are summarized in Appendix B.

## 2.2 The retrieved state

The relationship described by Eq. (6) requires information about particle size distributions and single-scattering properties. With scattering properties and their uncertainties specified a priori as described in section 2.1.1, this leaves the snow PSD parameters and their PDFs to be determined by the retrieval.

Snow PSDs are frequently characterized as exponential

$$N(D) = N_0 \exp(-\lambda D) \tag{9}$$

where $\lambda$ is the slope of the distribution and $N_0$ its intercept. Rogers (1973) used photographs of snowflakes to develop estimates of snow size distributions based on actual dimensions and found snow size distributions to be exponential. Brandes et al. (2007) evaluated both exponential and gamma forms, which have the ability to represent sub- or super-exponential behavior, for snow

size distributions observed by a 2D video disdrometer over the course of several winter seasons. Although about 22% of the observed snow distributions exhibited super-exponential features, more commonly the fitted gamma distributions were nearly equivalent to exponential distributions. Several aircraft-based studies using in situ observations under a wide range of atmospheric conditions have confirmed exponential behavior, especially at larger particle sizes (Passarelli, 1978; Houze et al., 1979; Lo and Passarelli, 1982; Gordon and Marwitz, 1984; Braham, 1990; Woods et al., 2008) . While other studies of aircraft

observations have noted departures from exponential behavior (e.g., "super-" or "sub-exponential", Herzegh and Hobbs, 1985), Heymsfield et al. (2008) examined the suitability of exponential distributions for snow. They found that fitted exponential distributions, when used to simulate IWC and Ze, could provide generally good agreement with IWC and Ze calculated directly from the observed discrete size distributions. These studies support the adequacy of exponential distributions for retrieving snowfall. $D$ may be an actual dimension of the snow particle, the diameter of an equivalent mass ice sphere, or the melted drop

diameter. The choice is significant because $N_0$ and $\lambda$ depend on the choice of $D$. For this work, we use the maximum particle dimension, $D_M$, because $D_M$ is closely related to the particle dimensions measured by imagers such as video disdrometers (Wood et al., 2013) and aircraft particle probes, making comparisons with other datasets more straightforward.

The exponential size distribution parameters $N_0$ and $\lambda$ are the desired state variables. Values for $N_0$ may range over several orders of magnitude, so $\log(N_0)$ is retrieved instead. The variability of $\lambda$ is significantly smaller than that of $N_0$; however,

examination of fitted exponential distributions from C3VP snow events indicated that the distribution of values for $\lambda$ was strongly non-Gaussian. The log-transformed values are much less skewed (Fig. 1a), and accordingly, $\log(\lambda)$ is retrieved instead





. The corresponding state vector to be retrieved is then

$$\hat{\boldsymbol{x}} = \left[ \begin{array}{c} \log(N_0) \\ \log(\lambda) \end{array} \right],$$ (10)

and the associated covariance matrix obtained from the retrieval is of the form

$$\hat{\mathbf{S}}_x = \left[ \begin{array}{cc} s^2(\log(N_0)) & s(\log(N_0),\\ & \log(\lambda)) \\ s(\log(N_0), & s^2(\log(\lambda)) \\ \log(\lambda)) & \end{array} \right].$$ (11)

## 2.3 Prior estimates of the state

For each profile, the a priori state consists of a vector of expected values $\boldsymbol{x}_a$ and the corresponding covariance matrix $\mathbf{S}_a$, having the same sizes as the state vector $\boldsymbol{x}$ (Eq. (10)) and its covariance matrix $\mathbf{S}_x$ (Eq. (11)). A priori estimates of $\log(N_0)$ and $\log(\lambda)$ are determined using temperature-based parameterizations derived using snow PSDs observed during C3VP and

other field experiments. Exponential size distributions were fit to the observed size spectra from both ground-based Snowflake Video Imager, or SVI (Newman et al., 2009; Wood et al., 2013), and from 2D particle probes carried aboard the National Research Council Canada's Convair-580 during three C3VP research flights (Fig. 1b). Results from a number of earlier studies are shown as well for comparison, including ground-based observations taken in and near the Rocky Mountain Front Range (Rogers, 1973; Brandes et al., 2007); and aircraft observations over the central Sierra Nevada (Gordon and Marwitz, 1984,

1986), in lake effect snow over Lake Michigan (Braham, 1990), in synoptic snowfall over central Illinois (Passarelli, 1978), and in orographic and frontal wintertime precipitation in the Pacific Northwest (Woods et al., 2008). Also shown are similar fits performed on 2D probe observations from a Wakasa Bay research flight on 27 January 2003 (Lobl et al., 2007). The results suggest that the C3VP observations adequately represent snowfall from a number of different regimes, although the number concentrations from several studies are at the margins of the C3VP observations.

Both $\lambda$ and $N_0$ have been observed to vary log-linearly with temperature (e.g., Houze et al., 1979; Woods et al., 2008; and works reviewed in Ryan, 1996). Fits were therefore constructed for both parameters using the combined C3VP aircraft and SVI data and uncertainties estimated using residual standard deviations (RSDs) calculated for data binned into 2 K intervals (Fig. 2). The narrow temperature ranges for the Wakasa Bay and Brandes et al. (2007) observations make comparisons against the C3VP temperature dependence uninformative. For $\lambda$, the Rogers (1973) observations are largely outside the bounds of

the RSDs, but are generally consistent with the C3VP histogram for warmer temperatures. The aircraft observations other than Wakasa Bay follow a temperature trend similar to the C3VP observations. For $N_0$, several of the comparison datasets lie mostly above the RSD bounds, but would be well within a +2 RSD bound.

Based on the similarity of C3VP with results from other experiments, the a priori states derived from these observations can be expected to represent a broad range of snowfall regimes and were adopted for the retrieval. A priori values for $\log(\lambda)$ and



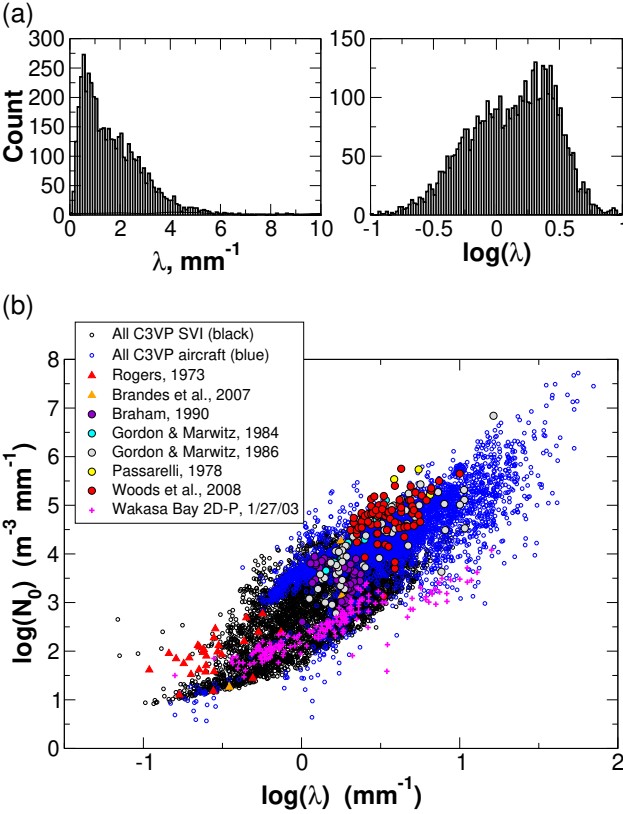

**Figure 1.** a) Histograms of $\lambda$ and $\log(\lambda)$ fitted to C3VP SVI observations. b) Estimates of $\lambda$ and $N_0$ determined from fits to size distributions from C3VP observations, with values provided from several earlier studies for comparison.

$\log(N_0)$ were estimated from the linear fits as

$$\log(\lambda_{ap}) = -0.03053(T - 273.) - 0.08258 \tag{12}$$

$$\log(N_{0,ap}) = -0.07193(T - 273.) + 2.665$$

with $\lambda$ in mm$^{-1}$, $N_0$ in m$^{-3}$ mm$^{-1}$, and T in K. The RSDs show little variation with temperature except in the vicinity of 240 K, where they increase substantially. These large RSDs are in response to a few outlying samples with small $\lambda$ and $N_0$ values.

Accordingly, variances were treated as constant and were estimated as the squared RSDs averaged over all temperatures. The uncertainty model also includes the covariance between $\log(N_0)$ and $\log(\lambda)$. Correlation coefficients were evaluated for each of the temperature-binned data subsets. giving a mean coefficient of 0.72 with a standard deviation of 0.12. The a priori covariance was modeled as $0.72 \cdot s\left(\log(\lambda_{ap})\right) \cdot s\left(\log(N_{0,ap})\right)$ :





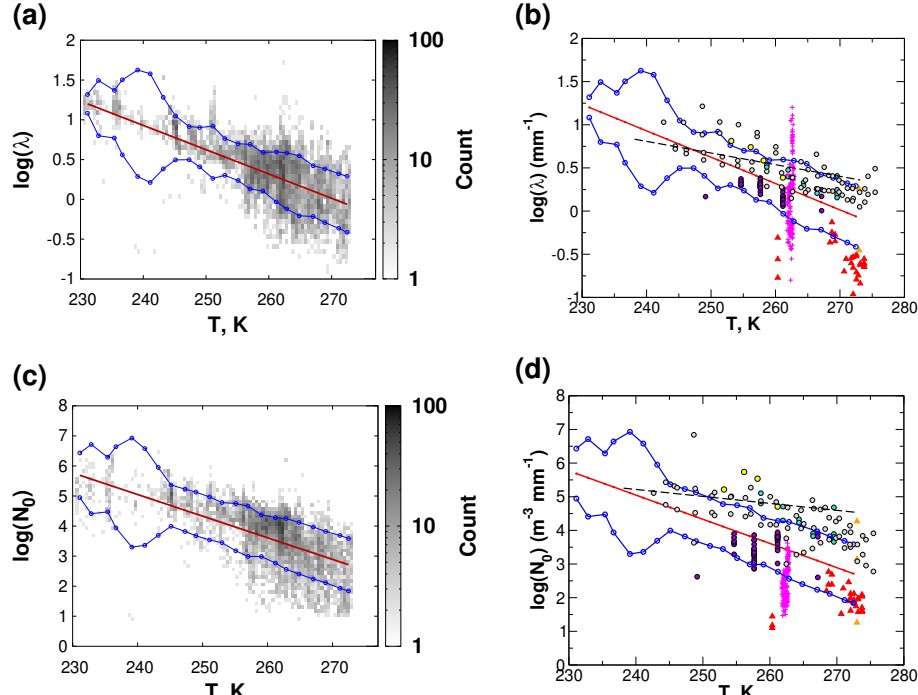

**Figure 2.** Dependence of $\log(\lambda)$ and $\log(N_0)$ on temperature. Central red lines show the best-fit relationships, while the upper and lower blue lines show bounds given by +/- 1 residual standard deviation. The shaded gray shows the 2D histogram of values for the C3VP surface and aircraft observations (panels a and c). Symbols (panels b and d) match those from Figure 1 except that, in lieu of symbols for Woods et al. (2008), the dashed black line shows a linear best fit reported by the authors.

$$
\begin{aligned}
s^2\left(\log\left(\lambda_{ap}\right)\right) &= 0.133 \\
s^2\left(\log\left(N_{0,ap}\right)\right) &= 0.95 \\
s\left(\log\left(\lambda_{ap}\right),\log\left(N_{0,ap}\right)\right) &= 0.26
\end{aligned}
\tag{13}
$$


## 3 Implementation and uncertainty sources

Applying the exponential distribution in Eq. (6), the singly-scattered nonattenuated reflectivity $Z_e^{ss}$ is

$$
Z_e^{ss}(R) = \frac{\Lambda^4}{\|K_w\|^2 \pi^5} \int\limits_{D_{M,min}}^{D_{M,max}} N_0 \exp\left(-\lambda D_M\right) \sigma_{bk}(D_M,\tilde{\boldsymbol{b}}) \, dD_M.
\tag{14}
$$

The backscatter cross-section $\sigma_{bk}$ has been written to show its dependence on a vector of parameters $\tilde{\boldsymbol{b}}$ as well as on $D_M$. The vector $\tilde{\boldsymbol{b}}$ includes the parameters for the mass- and area-dimension relations $\alpha$, $\beta$, $\gamma$, and $\sigma$ which were used to construct the par-





ticle models from which the scattering properties were calculated. The tilde indicates that these parameters are approximations of the true values and a source of uncertainty.

## 3.1 Model-measurement uncertainties

The error covariance matrix $\mathbf{S}_\epsilon$ is

$$
\begin{aligned}
\mathbf{S}_\epsilon &= \mathbf{S}_y + \mathbf{S}_F \\
&= \mathbf{S}_y + \mathbf{S}_B^{ss} + \mathbf{S}_F^{ss}
\end{aligned}
\tag{15}
$$

where $\mathbf{S}_y$ is the covariance matrix for the measurement uncertainties and $\mathbf{S}F$ is that for the singly-scattered reflectivities given in Eq. (14). The forward-model uncertainties may be further decomposed as the sum of two terms: $\mathbf{S}_B^{ss}$, which is a covariance matrix describing uncertainties due to the forward model parameters $\tilde{b}$, and $\mathbf{S}_F^{ss}$, which is a covariance matrix describing uncertainties due to other assumptions in the calculation of $Z_e^{ss}$.

### 3.1.1 Uncertainties for measured reflectivities

The sources of reflectivity measurement uncertainty $\mathbf{S}_y$ include uncertainty in the absolute radiometric calibration and measurement noise. For this work, we assume the radar is well-calibrated, leaving noise as the uncertainty source, and model noise

using the well-characterized CloudSat CPR (Tanelli et al., 2008). For reflectivities above -10 dBZ, one standard deviation of noise as a fraction of the mean signal is about -16 dB, while for reflectivities below -10 dBZ, noise is an increasing fraction of the signal, reaching 0 dB at the minimum detectable signal of -30 dBZ (R. Austin, personal communication, 4 November, 2008). The resulting uncertainties range from 3 dBZ for a reflectivity of -30 dBZ to about 0.1 dBZ for reflectivities above -10 dBZ$_e$. (Fig. 3).

### 3.1.2 Forward model uncertainties

Uncertainties $\mathbf{S}_B^{ss}$ due to the forward model parameters $\tilde{b} = (\alpha, \beta, \gamma, \sigma)^{\mathrm{T}}$ that describe the snow particle model were examined in Wood et al. (2015) as

$$
\mathbf{S}_B^{ss} = \mathbf{K}_b \mathbf{S}_b \mathbf{K}_b^{\mathsf{T}}
\tag{16}
$$

where $\mathbf{K}_b$ is the Jacobian of the forward model reflectivities with respect to the parameters $\tilde{b}$ and $\mathbf{S}_b$ is the covariance matrix

for the parameters. $\mathbf{K}_b$ depends on the estimated state $\hat{x}$ and so is evaluated at each iterative step via finite differences using a set of perturbed particle models. Wood et al. (2015) found these uncertainties to be near 5 dB, increasing to as high as 15 dB for very broad distributions.

$\mathbf{S}_F^{ss}$ quantifies uncertainties due to other assumptions and limitations in the forward model reflectivity calculation. Wood et al. (2015) looked at uncertainties due to the random component of dipole placement within DDA models for a particular particle



Atmospheric
Measurement
Techniques

Discussions

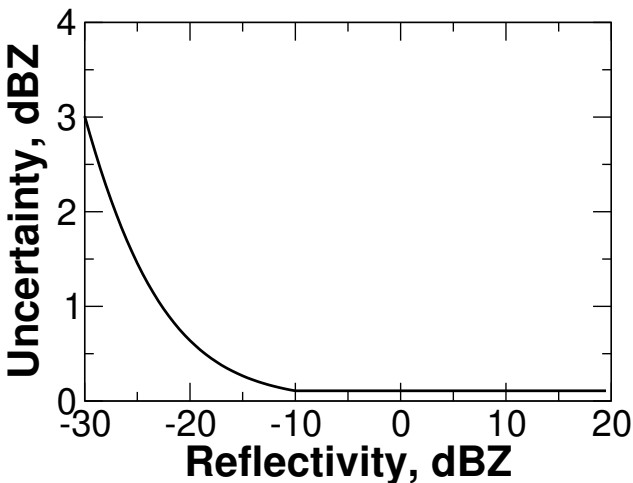

**Figure 3.** Uncertainty based on one standard deviation of noise for the CloudSat CPR.

shape and found them negligible. Other sources include the assumption of the shape of the distribution as exponential, the choice of particle shape, and the discretization and truncation of the integrations over size distribution.

Errors due to the assumed exponential shape were evaluated using a dataset of 4080 SVI-measured, discrete, 5-minute-duration snow PSDs from C3VP. Simulated reflectivities and snowfall rates were calculated using the B8pr-30 particle model and the Mitchell and Heymsfield (2005) terminal velocity model. Exponential distributions were fit to the observed discrete PSDs using orthogonal distance regression (Boggs et al., 1992; Jones et al., 2001) with uncertainty estimates per Wood et al. (2013). The fitted distributions were scaled in number concentration to match the snowfall rates simulated from the discrete distributions. The fitted distributions were then used to simulate reflectivities for comparison against those from the discrete distributions. Errors are negligible at high reflectivities but increase as reflectivity decreases (Fig. 4). Bias is negligible, and the total uncertainty is modeled as

$$s_F^2(dB) = \left[\exp\left(-\left(dBZe + 14\right)/16\right)\right]^2, \tag{17}$$

reaching a maximum of 1 dB of uncertainty at -15 dBZe.

Uncertainties due to shape were evaluated using the same SVI dataset to which the alternate particle shapes Ep (ellipsoidal) and B8pr-45 (branched spatial particle with larger aspect ratio than B8pr-30) from Wood et al. (2015) were applied to simulate reflectivities. These alternate shapes are constrained to have the same mass-dimension relationship as used for the B8pr-30 particle model used in this work, so differences are due only to particle shape. Figure 5 shows total and variance-only RMS errors. From these results we estimate the shape uncertainty to be 2 dB.

Finally, truncation and discretization errors were evaluated using the same SVI PSD dataset. These are errors that result from the discrete treatment of the integrations over size distribution, errors due to both the limited maximum $D_M$ in the particle model and in the limited resolution of the particle model. Truncation errors were evaluated using analytic exponential PSDs fitted to the SVI PSD dataset as described previously. The particle model backscatter properties were augmented to

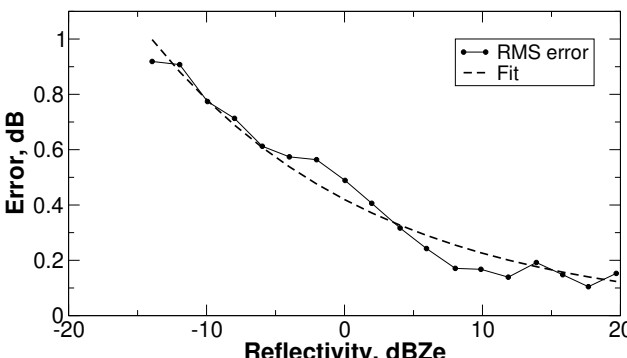

**Figure 4.** Actual RMS errors and the fitted model for uncertainty due to the assumed exponential size distribution.

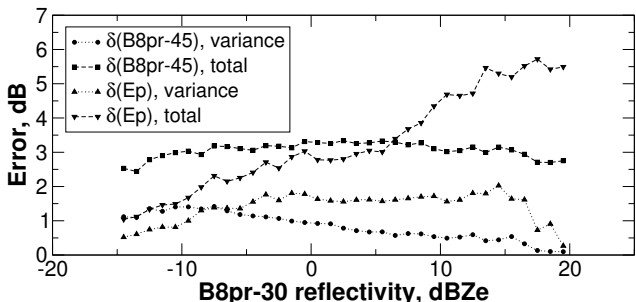

**Figure 5.** Errors in reflectivity for the Ep and B8pr-45 shapes compared to the B8pr-30 shape. Errors shown are total (bias + variance) and variance only.

$D_M = 40\,\mathrm{mm}$ by linearly extrapolating backscatter efficiencies, then reflectivities were calculated using integrations to both the standard (maximum $D_M = 18\,\mathrm{mm}$) and augmented size range. The bias and scatter of the truncation errors were -0.1 and 0.42 dB. To evaluate discretization errors, a high-resolution version of the particle model backscatter properties was created by interpolating backscatter efficiencies so that the particle size resolution of the particle model was increased by a factor of two. Reflectivities were then calculated and compared against those from the standard-resolution particle model. The bias and scatter of the discretization errors were 0. and 0.02 dB.

### 3.2 Snowfall rate and uncertainties

The snowfall rate $P$ in units of liquid water depth per unit time is

$$P(R) = \frac{1}{\rho_{liq}} \int_{D_{M,min}}^{D_{M,max}} N(D_M, R) m(D_M, R) V(D_M, R) \, dD_M, \tag{18}$$

where $m(D_M, R)$ is particle mass, $V(D_M, R)$ is fallspeed, and $\rho_{liq}$ is the density of liquid water. Particle mass is provided by Eq. (7). Fallspeed is assumed to equal terminal velocity, which is calculated from the model of Mitchell and Heymsfield





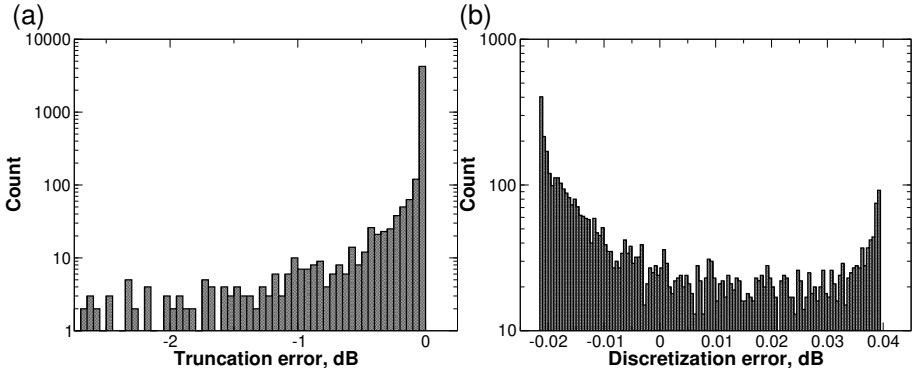

**Figure 6.** Histograms of errors for truncation and discretization.

(2005) using particle mass, the horizontally-projected area from Eq. (8), and environmental pressure and temperature from collocated observations. Uncertainties for the estimated snowfall rate are determined in a manner similar to that used for the forward model uncertainties. The total variance $\mathbf{S}_P$ is decomposed as

$$\mathbf{S}_p = \mathbf{S}_{\hat{x}} + \mathbf{S}_{\tilde{b}} + \mathbf{S}_v + \mathbf{S}_{exp} \tag{19}$$

where the terms on the right represent the variances resulting from 1) retrieved state uncertainties, 2) particle model parameter uncertainties, 3) uncertainties in the fallspeed model and its parameters, and 4) assuming an exponential form for the PSD, respectively. Contributions from uncertainties in the retrieved state and in the particle model parameters are determined using linearized error propagation (e.g., following a form like Eq. (16)). Fallspeed contributions are handled following Wood et al.

(2014). Snowfall rate uncertainties due to the assumed exponential form of the size distribution are determined using the SVI PSD dataset in an approach analogous to that for Eq. (17). In this approach, number concentrations for the fitted exponential distributions were scaled so that reflectivities were matched, then snowfall rate errors were evaluated. The fractional uncertainty in snowfall rate was found to be

$$f_P = -0.06 \log(P) + 0.05 \tag{20}$$

from which the necessary variance can be determined. Uncertainties from each of the four sources are treated as uncorrelated.

## 4    Retrieval performance tests with ground-based radar observations

During C3VP, a vertically-pointing W-band radar (the Jet Propulsion Laboratory's Airborne Cloud Radar, ACR) was deployed on the ground at CARE. In all, about 28 hours of ACR radar profiles of snowfall were recorded at approximately 2.8 s intervals. These observations represent 17 distinct snow events that occurred over 18 days between 3 November 2006 and 2 March 2007;

however, most of the accumulations were concentrated during nine of the events (Table 1). The data include three of the cases used to develop the snow particle microphysical and scattering models (Wood et al., 2015). Of the nearly 36,000 ACR profiles, approximately 7300 are from these three cases.





**Table 1.** Accumulations by event for the ACR retrievals. Two distinct events, indicated as (a) and (b), occurred on 20 Jan 2007. Duration shows the elapsed time of ACR observations for which retrievals were performed. Fractional differences are relative to FD12P accumulations.
[*]Accumulations adjusted to remove anomalies indicated in Figure 7.

| Date | Duration | Accumulations | | |
| | | ACR | FD12P | Fractional |
| | h | mm LWE | | difference, % |
| --- | --- | --- | --- | --- |
| 3 Nov 2006 | 0.98 | 0.065 | 0.11 | -40.9% |
| 2 Dec 2006 | 0.16 | 0.007 | 0.00 | — |
| 6 Dec 2006 | 4.00 | 0.86 | 0.80 | 7.5% |
| 7 Dec 2006 | 1.08 | 0.038 | 0.093 | -59.1% |
| 8 Dec 2006 | 0.34 | 0.018 | 0.00 | — |
| 17 Jan 2007 | 0.09 | 9.3e-04 | 0.00 | — |
| 19 Jan 2007 | 0.46 | 0.061 | 0.13 | -53.1% |
| 20 Jan 2007 (a) | 0.32 | 0.004 | 2.8e-04 | 1329% |
| 20 Jan 2007 (b) | 0.59 | 0.079 | 0.0 | — |
| 22 Jan 2007 | 4.29 | 0.89 | 0.87 | 2.2% |
| 23 Jan 2007 | 0.76 | 0.017 | 0.00 | — |
| 26 Jan 2007 | 0.93 | 0.045 | 0.085 | -47.1% |
| 27 Jan 2007 | 3.36 | 0.57 | 1.06 | -46.2% |
| 19 Feb 2007 | 0.97 | 0.26 | 0.18 | 44.4% |
| 22 Feb 2007 | 2.72 | 0.40[*] | 0.23[*] | 73.9% |
| 26 Feb 2007 | 2.41 | 0.58 | 0.64 | 9.4% |
| 1 Mar 2007 | 4.23 | 1.14[*] | 1.57[*] | -27.4% |
| Season | 26.3 | 5.04[*] | 5.77[*] | -12.6% |

The retrieval was applied to the ACR reflectivities observed in the single range bin nearest the surface, at 197 m above ground level (AGL). Temperatures and pressures needed by the retrieval to perform snow detection, calculate fallspeeds and

establish the a priori states were obtained from nearby surface meteorology observations. Because of the short distance to the target range bin, attenuation along the path was neglected. For comparisons, snowfall rate observations were obtained at 1-minute intervals from the Vaisala FD12P (Viasala Oyj, 2002) and scaled to provide unbiased accumulations relative to the nearby Dual Fence Intercomparison Reference, or DFIR (Goodison et al., 1998). The retrieved ACR snowfall rates, $P_{ACR}$, were matched to the nearest-in-time observed snowfall rate, $P_{FD12P}$.

Time series of $P_{ACR}$ and $P_{FD12P}$ show a high degree of agreement over most of the observing period (Fig. 7, upper panel). Two notable exceptions occur near time indices 25000 and 32500, when the FD12P recorded snowfall rates above 1 mm LWE h[-1] while the retrieved values are substantially smaller. Examining the time series of ACR reflectivities shows that the ACR did not observe high reflectivities during these periods (Fig. 7, lower panel). The first of these anomalies occurred 22

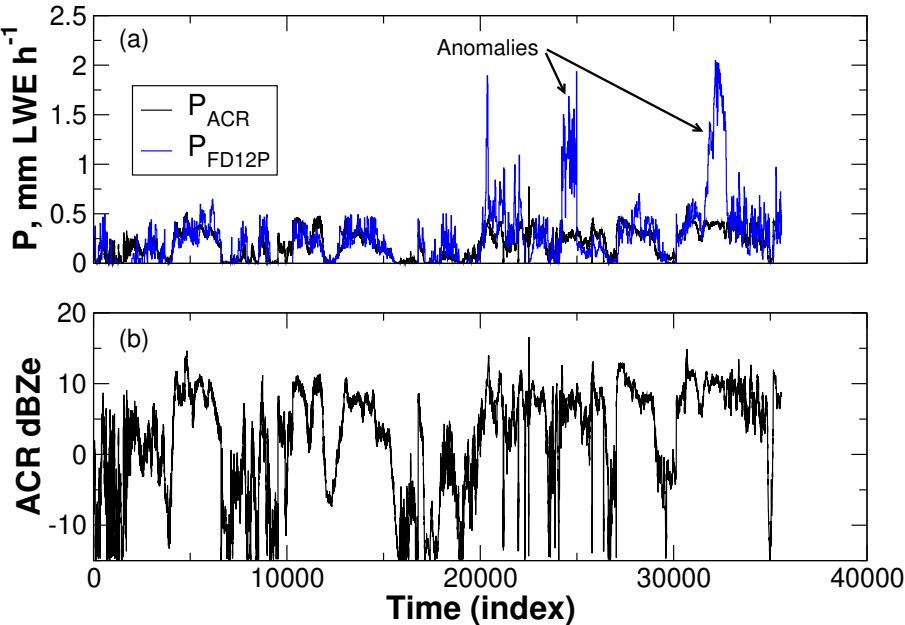

**Figure 7.** Upper panel: Time series of snowfall rates retrieved from ACR reflectivities and observed. Lower panel: Corresponding time series of ACR reflectivities. Each time index indicates a 2.8 s observation by the ACR. Snowfall rates retrieved for the ACR used the reflectivity in the range bin nearest the surface, at 197 m AGL.

February 2007 from 11:20 to 12:05 UTC while the second occurred 1 March 2007 between 22:15 and 22:50 UTC. For both, the
ACR operator made note of the heavy snowfall, suggesting that both the FD12P and the ACR observed similar snowfall rates (Austin et al., 2007). Based on soundings, Environment Canada forecasts, and ACR operator observations, these anomalies appear to correspond with melting aloft, ice pellets, and freezing rain (Wood, 2011), which violate the retrieval's assumption of dry, aggregate-like snow particles.

Accumulations were calculated from both $P_{ACR}$ and $P_{FD12P}$ with and without the two anomalies described above (Fig. 8).
Accumulations agree substantially during the first 16 hours but diverge somewhat beyond that. With the anomalies included the final difference between the accumulations is 2 mm. With the anomalies removed that difference is reduced to 0.7 mm. For individual events, absolute fractional differences between the ACR and FD12P accumulations can range to 50% and upwards (Table 1), but these large values are associated mainly with events with small accumulations. For events with larger accumulations, the absolute fractional differences are mostly below 30%. At seasonal timescales, the random components in
event-total accumulations are likely uncorrelated, leading to offsetting errors when calculating seasonal accumulations. The time series of absolute fractional differences between the ACR-derived and FD12P accumulations begins with large fractional differences. Within 5 hours and over the initial three events, the fractional differences reduce to less than 5%, then remain below 20% for the remainder of the season.



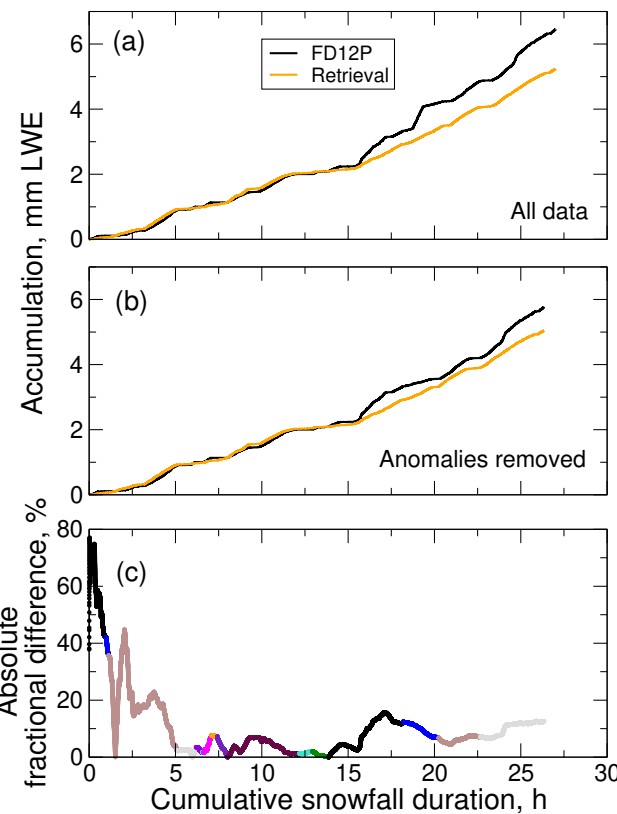

**Figure 8.** Snow accumulations computed from $P_{ACR}$ and $P_{FD12P}$. The accumulations are for 17 snow events observed by the ACR on 18 days between 3 November 2006 and 2 March 2007, but accumulations are principally from nine events (Table 1). The events were concatenated sequentially in time and the time axis indicates the cumulative time over all events. (a) Accumulations from all observations and corresponding retrieval results, (b) accumulations with two anomalous periods identified in Figure 7 removed, and (c) fractional differences in accumulations shown in panel (b), with distinct colors indicating individual events.

## 4.1 Snowfall rate uncertainties

Uncertainties in instantaneous snowfall rate estimates, taken to be the square root of the total variance evaluated as shown in Eq. (19), were evaluated by binning the fractional uncertainties by snowfall rate then averaging and taking standard deviations. Mean fractional uncertainties range from 150% to 185%, and the range for +/- 1 standard deviation extends from about 145% to 190% (Fig. 9). The fractional uncertainties generally increase with increasing snowfall rate, but above 0.5 mm LWE h$^{-1}$ the means and standard deviations diminish and result from only a small number of samples in each bin. For comparison,

uncertainties for FD12P precipitation rates at 5-minute resolution were estimated at 0.03 mm h$^{-1}$ for rates less than 0.05 mm h$^{-1}$, 50% for rates up to 0.5 mm unith$^{-1}$, and 30% for rates larger than 0.5 mm h$^{-1}$ by Wood et al. (2014) based on comparisons against a Precipitatation Occurrence Sensor System.



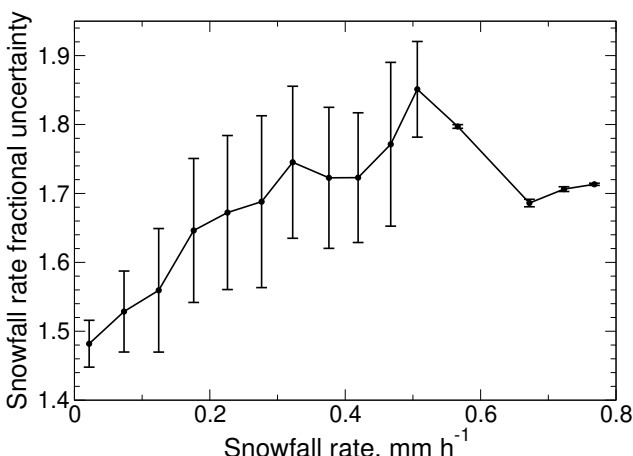

**Figure 9.** Instantaneous fractional uncertainties in snowfall rate. The central line shows mean fractional uncertainties and the error bars show +/- 1 standard deviation.

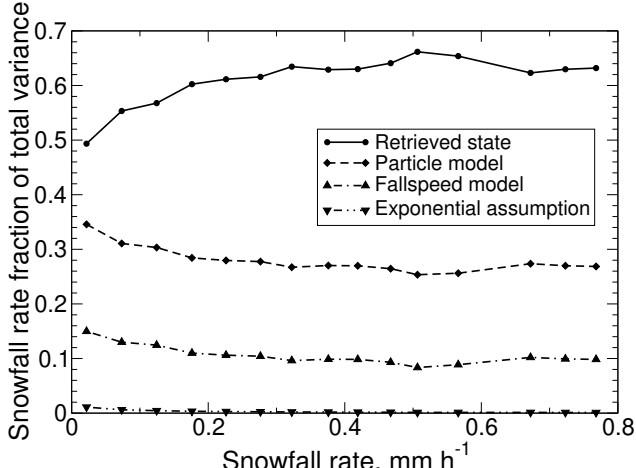

**Figure 10.** Instantaneous fractional variances for snowfall rate resolved by source.

To evaluate the importance of each source of uncertainty, variances from each of the sources from Eq. (19); (retrieved state, microphysical parameters, fallspeed parameterization, or exponential distribution) were extracted separately, then fractions of total variance were calculated. To allow the trends in each source to be shown as a function of snowfall rate (Fig. 10), the fractions were binned by snowfall rate and averaged. As snowfall rates increase up to 0.5 mm h$^{-1}$, the variance due to the retrieved state becomes a more significant contributor to the total variance, while the contributions from the other sources diminish. The contribution due to the assumed exponential PSD shape is not significant.

The instantaneous uncertainties for snowfall rate include uncertainties due to random errors and biases in the retrieval components and observations. For accumulations or mean rates evaluated over longer time periods, errors due to random sources



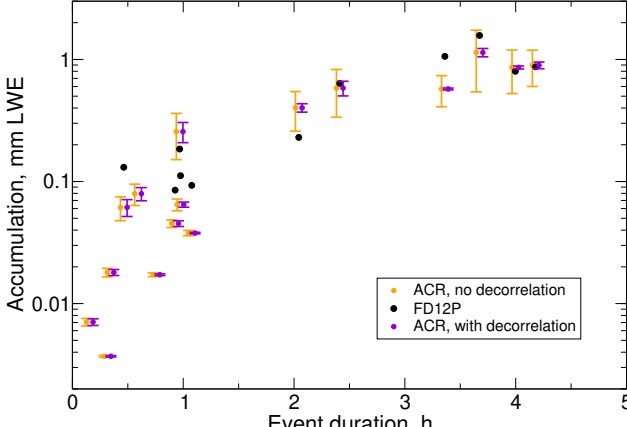

**Figure 11.** End-of-event accumulations and uncertainties. The $P_{ACR}$ accumulation uncertainties are estimated assuming intra-event errors are perfectly correlated (orange) and decorrelated using a negative exponential model with a decorrelation scale of 0.5 hours (purple). $P_{FD12P}$ accumulations (black) are shown for comparison except for those equal to zero which are omitted. For clarity, the $P_{ACR}$ accumulations are plotted at +/- 0.02 hours (purple/orange) of their actual durations.

may be reduced and remaining errors can be more representative of biases in the retrieval. The reductions in random errors depend on their correlations in time, however (e.g., Taylor, 1997). When random errors within events are assumed perfectly positively correlated, end-of-event $P_{ACR}$ accumulations have fractional uncertainties from 1.5% to 52.4% (Fig. 11). In actuality, the random error sources likely decorrelate with increasing separation in time. While the scales for these decorrelations are

not known, with even a modest amount of decorrelation in the errors the uncertainties are reduced substantially. After applying a negative exponential decorrelation model with a decorrelation scale of 0.5 hour to intra-event errors, the fractional uncertainties at the ends of individual events are 1.3% to 18.8%. The most significant reductions due to decorrelation occur with the longer-duration events. The end-of-season $P_{ACR}$ accumulation uncertainties, calculated assuming inter-event uncertainties are uncorrelated, are reduced from 64.9% for perfectly correlated to 11.8% for decorrelated intra-event errors.

Agreement between observed $P_{FD12P}$ event accumulations and those from $P_{ACR}$ generally improves for events with larger accumulations and durations (Fig.11 ). Of the seven events with accumulations larger than 0.2 mm and durations of 1 h and longer, the $P_{FD12P}$ accumulations for six fall within or near the uncertainty bounds of the $P_{ACR}$ accumulations with perfectly correlated errors, while four out of seven are within or near the much narrower bounds for errors with decorrelations. This result is also true for the season as a whole. For the duration of 26.3 hours and accumulation of 5.05 mm from $P_{ACR}$, the difference

compared to the $P_{FD12P}$ seasonal accumulation of 5.77 mm is -12.6%. The difference is similar to the $P_{ACR}$ accumulation uncertainty of 11.7% for decorrelated errors.





## 4.2 Information content

The optimal estimation results allow easy calculation of a number of metrics that quantify retrieval performance in terms of information content (Rodgers, 2000; Shannon and Weaver, 1949). These include the averaging kernel matrix


$$\mathbf{A} = \left( \hat{\mathbf{K}}^{\mathsf{T}} \mathbf{S}_{\epsilon}^{-1} \hat{\mathbf{K}} + \mathbf{S}_a^{-1} \right)^{-1} \hat{\mathbf{K}}^{\mathsf{T}} \mathbf{S}_{\epsilon}^{-1} \hat{\mathbf{K}}, \tag{21}$$

the Shannon Information Content

$$H = \frac{1}{2} \log_2 \left| \mathbf{S}_a \hat{\mathbf{S}}_x^{-1} \right|, \tag{22}$$

and the degrees of freedom for signal

$$d_S = \mathrm{Tr}\left( \mathbf{A} \right). \tag{23}$$

Briefly, the diagonal values of $\mathbf{A}$ indicate the degree to which the corresponding retrieved state variables are determined by the observations (values nearer 1) versus the a priori (values nearer 0). $H$ measures how well the observations serve to narrow the possible retrieved states in comparison to the a priori. Its value can be interpreted as describing the binary bits of resolution of the observing system (L'Ecuyer et al., 2006). $d_S$ quantifies the number of independent quantities that are determined by the observations. See Rodgers (2000) for a more complete discussion in the context of retrieval theory.

For the ACR retrievals, values for $H$ vary between 0.4 and 1.2 (Fig. 12), indicating that the measurements resolve between 1.3 and 2.3 distinct states. Values for $d_s$ show that the retrieval produces somewhat less than one independent piece of information that is significant compared to the measurement and forward model uncertainties. The lower two panels of Figure 12 show the diagonal elements of $\mathbf{A}$. While the element relevant to $\lambda$, $\mathbf{A}\left[\log\left(\lambda\right)\right]$, is consistently positive, the element for $N_o$ , $\mathbf{A}\left[\log\left(N_0\right)\right]$ is near zero and is at times negative. These results show that $\log\left(\lambda\right)$ is moderately to strongly constrained by the

reflectivity observation, while $\log\left(N_0\right)$ is largely dependent on the a priori constraint.

The shape of the size distribution plays a significant role in determining the values of these metrics. Information content $H$ increases as the distribution narrows (Fig. 13, panel a). The increase in $H$ accompanies a substantial increase in the magnitude of the sensitivity of the forward model to $\log\left(\lambda\right)$ (panel b). This increased sensitivity allows the observed reflectivity to better constrain the retrieved state, particularly the value of $\log\left(\lambda\right)$. As a result, $\mathbf{A}\left[\log\left(\lambda\right)\right]$ increases from 0.4 to 0.95 as $\lambda$ increases

(panel c). The behavior of $\mathbf{A}\left[\log\left(N_0\right)\right]$ (panel d) is quite different. The values are small and are positive for small values of $\lambda$, but become negative as $\lambda$ increases. This behavior results from the positive a priori correlation between $\log\left(\lambda\right)$ and $\log\left(N_0\right)$, and the opposing signs of the sensitivities of dBZ$_e$ to these two variables. While the forward model is strongly sensitive to $\log\left(\lambda\right)$, its sensitivity to $\log\left(N_0\right)$ is 3-4 times smaller in magnitude. This sensitivity has a constant value of 10 owing to the reflectivity in dBZ$_e$ being a linear function of $\log\left(N_0\right)$. Consequently, the retrieved value of $\log\left(\lambda\right)$ is influenced more strongly



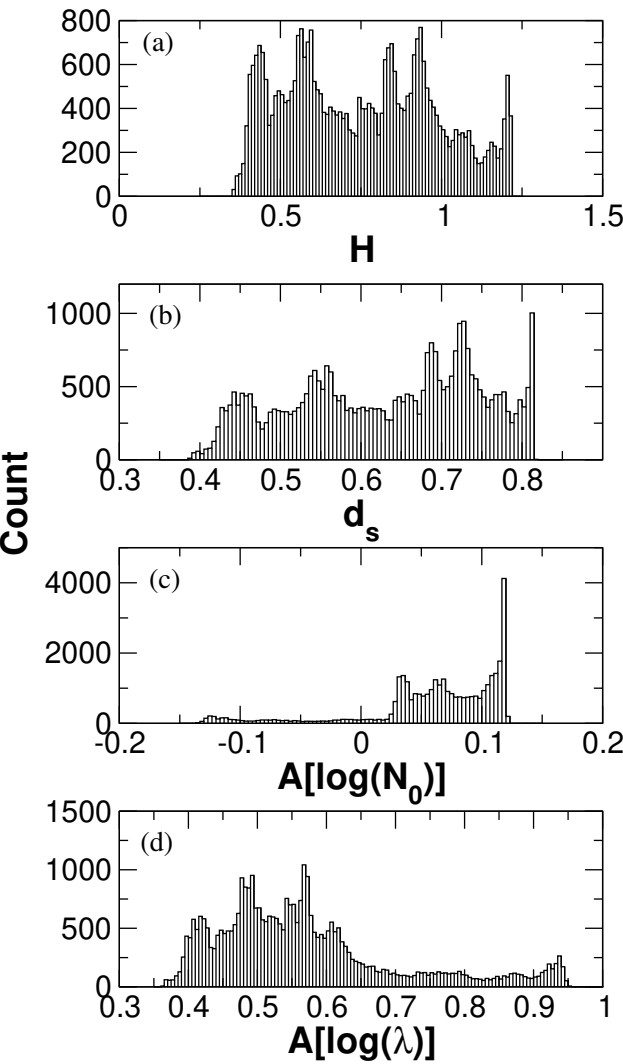

**Figure 12.** Distributions of information content metrics for the ACR retrieval.

by the observations, while the retrieved value of $\log(N_0)$ is influenced more by the a priori estimate of the state. This difference is reflected in panels (c) and (d) of Figure 13.

## 5 Discussion and conclusions

While millimeter-wavelength, single-frequency radar reflectivity observations alone would seem to have limited utility for retrieving snowfall properties, the results herein demonstrate capabilities for quantifying snowfall rate, accumulation and aspects of the snow PSD. The results were obtained by applying the radar observations to constrain a priori information appropriate to





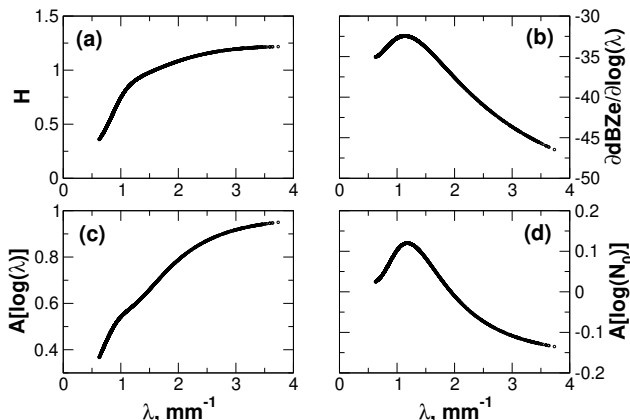

**Figure 13.** Information content metrics and the forward model Jacobian as functions of $\lambda$.

a broad range of snowfall regimes. The results indicate that the approach would provide useful information when applied to observations such as those from satellite-borne radars, which observe a range of snowfall regimes and for which radar observables are limited to reflectivity.

The results demonstrate the ability of the retrieval to produce reliable estimates of snow accumulation, particularly over time scales involving multiple events and more than several hours of snowfall duration, in spite of large uncertainties in retrieved instantaneous snowfall rates. For the C3VP season, the retrieval reproduced the observed accumulation within 13% at the end of the season. These results were achieved by omitting two particular time periods during which the retrieval's dry snow assumptions were inconsistent with the observed snowfall. Without this adjustment, the end-of-season absolute difference was 18.9%, illustrating the need for adequate discrimination of the precipitation phase in the retrieval process. The time series of seasonal accumulation shows that while the initial fractional differences reach almost 80%, the differences diminish with time and increasing accumulation, reaching values of less than 5% within five hours. For individual events, best agreement between the observed and retrieved snow accumulations were achieved for events that were longer in duration and produced more substantial accumulations. The observed accumulations for these events were mostly near or within the tighter uncertainty bounds produced by a decorrelating error model applied to the retrieved accumulation. The modest decorrelation used in the model produces uncertainties in the retrieved event accumulations of only 1% to 20%. Thus despite large instantaneous snowfall rate uncertainties for these single-frequency, millimeter-wavelength retrievals, retrieved rates can be expected to prove of value for quantifiying accumulations over events, months, seasons, and longer.

Uncertainties in instantaneous retrieval-estimated snowfall rate are dominated by uncertainties in the retrieved state (the uncertainties in the estimated PSD), followed by uncertainties in particle model parameters and to a lesser extent, the uncertainties in the fallspeed model. Uncertainties due to the assumption of an exponential PSD form are negligible. There is a degree of ambiguity here. The uncertainties in the particle model parameters contribute to the uncertainties in the estimated snowfall rate due to the appearance of the mass term in Eq. (18) but also contribute to uncertainties in the retrieved state. We treat these as





**Table 2.** Contributions to uncertainties in forward-modeled and observed reflectivity.

| Source | Reflectivity, dB |
|---|---|
| Observed reflectivity | < 0.5 |
| Particle model | 5. - 15. |
| Shape | 2. |
| Assumed exponential | < 1. |
| Truncation | 0.42 |
| Discretization | 0. |
| Random dipole locations | 0. |

independent contributions to the uncertainty. There is likely some covariance that could reduce total uncertainties but this is not addressed in the treatment of snowfall rate uncertainty presented here.

Retrieval performance, quantified in terms of information content metrics, is determined by the sensitivity of the observations to the desired state vector, the uncertainties assessed for the forward models and measurements, and the explicit assumptions about the uncertainty in the a priori knowledge of the state. For W-band modeled reflectivities in dB, the magnitude of sensitivities for $\log(\lambda)$ are 3-4 times those for $\log(N_0)$, and sensitivities are opposite in sign. This contributes to $\log(\lambda)$ being better constrained by the retrieval than is $\log(N_0)$. The consequences of these sensitivities are described more fully in Appendix A.

To the extent that process information can be gleaned from changes in the slope parameter over time or space, the retrieval may be useful for process analyses when more direct observations of PSD are not available.

    Model-measurement uncertainties are dominated by uncertainties in the particle model parameters (e.g., the coefficients and exponents of the mass- and area-dimension relationships, Table 2), and it is the uncertainties in mass parameters that are the most substantial contributor (Wood et al., 2015). For these near-surface observations, contributions to uncertainties in W-

band radar reflectivity from shape, the assumption of an exponential form for the PSD, and the discrete-truncated form of the integrations over size distribution were not significant. For longer wavelength radars that might be used in similar applications (e. g., the MRR or KAZR), shape uncertainty will likely be even smaller due to less prevalent non-Rayleigh scattering.

    These baseline results suggest several avenues for improving such single-frequency, radar reflectivity-based snowfall retrievals. Improved constraints on snow PSD parameters, through either reduced a priori uncertainties or better observational

constraints, are paramount. For ground- or aircraft-based observations, ancillary measurements of snow PSDs can improve the a priori constraints. For retrievals from satellite-borne radar where such measurements are not available, the a priori state is given by more broadly applicable relationships for PSD parameters like those presented here. To the extent that a priori states for specific snowfall regimes might have smaller uncertainties, knowledge of regime-specific PDFs for snow PSD parameters would improve retrieval results provided the correct regime can be diagnosed by the retrieval. Coincident dual-frequency

radar observations may also provide improved constraints on the snow PSD parameters (Liao et al., 2005; Matrosov, 2011) but among current satellite-borne instruments, the CPR is single-frequency, and while the GPM DPR provides dual-frequency observations, the DPR sensitivities limit observations to heavier snowfall (Skofronick-Jackson et al., 2019) and implementation





of dual-frequency snowfall retrieval has proven difficult (Iguchi et al., 2018). Finally, model-measurement uncertainties can be reduced by reducing uncertainties in particle mass estimates. This may require a more synergistic approach in which improved

PSD information is coupled with additional observations such as Doppler velocity to better constrain the assumed particle model used in the retrieval, e.g., moving toward the approach used by Wood et al. (2014, 2015) with ground-based observations. The methods presented here, easily adaptable to other observing systems providing multiple frequency or collocated Doppler velocity observations, provide the basis from which such improvements can be tested and evaluated.

*Data availability.* Data used in this study can be obtained from the NASA Global Hydrology Resource Center's Distributed Active Archive

Center at https://ghrc.nsstc.nasa.gov under DOIs TBD.

## Appendix A: Retrieval interpretation

To interpret the behavior of the retrieval, we refer to the discussion of the information content metrics (Section 4.2). The small values for $\mathbf{A}\left[\log\left(N_0\right)\right]$ indicate its value is determined primarily by the a priori information and the negative signs do not fit the normal paradigm used to explain the $\mathbf{A}$ matrix. Their explanation reveals details of the significant behavior of this retrieval.

In the application of the retrieval to a single radar bin, the value of $\mathbf{A}\left[\log\left(N_0\right)\right]$ is given by

$$
\mathbf{A}\left[\log\left(N_0\right)\right] = \left[s^2\left(\log\left(\hat{N}_0\right)\right)\left(\frac{\partial dBZe}{\partial\log\left(N_0\right)}\right)^2 + \right.
$$
$$
\left. s\left(\log\left(\hat{N}_0\right),\log\left(\hat{\lambda}\right)\right)\left(\frac{\partial dBZe}{\partial\log\left(N_0\right)}\right)\left(\frac{\partial dBZe}{\partial\log\left(\lambda\right)}\right)\right]\left[s_y^2\left(dBZe\right)\right]^{-1}, \tag{A1}
$$

where the carets indicate retrieved values. In the first set of brackets on the right side, the sign of the first term is clearly positive, while that of the second term depends on the signs of the covariance and the two partial derivatives, which are the elements

of the Jacobian of the forward model. As was shown earlier (Fig. 13), $\frac{\partial dBZe}{\partial\log(N_0)}$ is positive while $\frac{\partial dBZe}{\partial\log(\lambda)}$ is negative. The covariance for the retrieved state changes very little from the a priori covariance, which is positive and represents a substantial correlation between $\log\left(\lambda\right)$ and $\log\left(N_0\right)$. This second term, then, is negative and as the magnitude of $\frac{\partial dBZe}{\partial\log(\lambda)}$ increases, the sign of $\mathbf{A}\left[\log\left(N_0\right)\right]$ changes from positive to negative.

These terms represent competing influences on the retrieved value of $\log\left(N_0\right)$. These competing influences arise from the

a priori covariance and from the Jacobian of the forward model. The positive covariance requires that a positive adjustment in $\log\left(\lambda\right)$ be accompanied by a positive adjustment in $\log\left(N_0\right)$. In contrast, the Jacobian terms have differing signs. If the difference between the observed and forward model reflectivity calls for a positive adjustment to $\log\left(\lambda\right)$, the corresponding adjustment to $\log\left(N_0\right)$ would be negative.

Figure A1 shows this process schematically. The size distribution that represents the initial state is shown by the solid line.

Assuming that the forward modeled reflectivity for this state overestimates the observed reflectivity (a positive error), two responses are possible: $\log\left(\lambda\right)$ could be increased, narrowing the distribution; and $\log\left(N_0\right)$ could be decreased, reducing the





amplitude of the distribution. Absent the covariance between $\log(\lambda)$ and $\log(N_0)$, the retrieval would apply both adjustments, likely giving more weight to the adjustment of $\log(\lambda)$ because of the stronger sensitivity of the forward model to that variable. These adjustments are represented by the heavy arrows labeled $\delta\log(N_0)$ and $\delta\log(\lambda)$. Because of the positive covariance between $\log(N_0)$ and $\log(\lambda)$, however, an increase in $\log(\lambda)$ produces an opposing response that increases $\log(N_0)$, shown by the upward-pointing heavy arrow. The resulting size distribution is shown by the dashed line.

For small $\lambda$ (broad distributions), the magnitude of $\frac{\partial dBZe}{\partial\log(\lambda)}$ is relatively small, so the covariance-driven adjustment is small and does not overcome the initial reduction in $\log(N_0)$. In these cases, $\log(N_0)$ decreases in response to a positive error in the modeled reflectivity. This net response is consistent with the sensitivity of the forward model to $\log(N_0)$ and $\mathbf{A}[\log(N_0)]$ is positive. For large $\lambda$ (narrower distributions), the magnitude of $\frac{\partial dBZe}{\partial\log(\lambda)}$ is larger. The covariance-driven adjustment is larger also and does overcome the initial reduction in $\log(N_0)$. As a result, $\log(N_0)$ increases in response to the positive error in the modeled reflectivity. Since this net response opposes the sensitivity of the forward model, $\mathbf{A}[\log(N_0)]$ is negative.

The combination of the strong positive covariance between $\log(N_0)$ and $\log(\lambda)$ and the comparatively weak sensitivity of the reflectivity to $\log(N_0)$ limits the behavior of the retrieval. For narrower distributions, the retrieval is prevented from simultaneously increasing $\log(\lambda)$ and decreasing $\log(N_0)$ in response to a positive error in reflectivity. The opposing behavior, decreasing $\log(\lambda)$ and increasing $\log(N_0)$ in response to a negative error in reflectivity, is also restricted. While correct in a climatological sense since $\log(\lambda)$ and $\log(N_0)$ are positively correlated, in nature there are likely scenes for which such responses would give a more accurate retrieval. This reasoning demonstrates how other measurements, specifically those with better sensitivity to $\log(N_0)$, would benefit the retrieval.

## Appendix B:  Particle model

The properties here are for the particle shape denoted as "B8pr-30" from Wood et al. (2015), an idealized 8-arm branched spatial particle. Values for the parameters of the mass- and area-dimension power functions are

$$\ln(\alpha) = -5.723$$
$$\beta = 2.248$$
$$\ln(\gamma) = -1.379$$
$$\sigma = 1.813$$

with error covariance matrix

$$\mathbf{S}_b = \begin{matrix} 0.592 & 0.212 & 0.090 & 0.023 \\ 0.212 & 0.142 & 0.011 & 0.007 \\ 0.090 & 0.011 & 0.335 & 0.103 \\ 0.023 & 0.007 & 0.103 & 0.046 \end{matrix}.$$





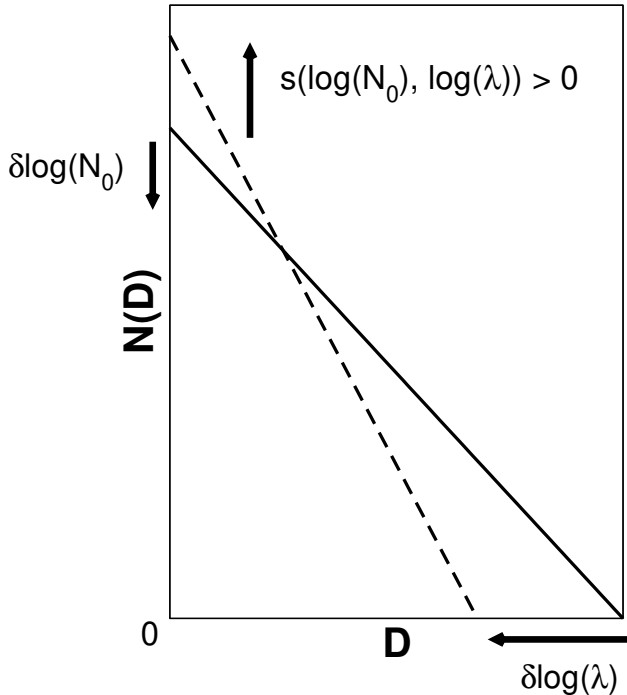

**Figure A1.** Schematic illustration of the retrieval process. The solid line represents the initial state of the retrieval while the dashed line shows the adjusted state assuming the initial state overestimates the observed reflectivity. The arrows labeled $\delta \log(\lambda)$ and $\delta \log(N_0)$ show the expected responses of the retrieval based on the sensitivities of the forward model. The arrow labeled $s\left(\log\left(N_o\right), \log(\lambda)\right)$ shows the response due to positive covariance between $\lambda$ and $N_0$.

These values are appropriate for use with particle size $D_M$ in centimeters, mass in grams and area in square centimeters. The
radar backscatter and extinction cross-sections are given in Table B1 versus particle size.

*Author contributions.* TSL and NBW developed the retrieval method from an initial concept by TSL. NBW performed the analyses and prepared the manuscript with contributions from TSL.

*Competing interests.* The authors declare that they have no conflict of interest.

*Acknowledgements.* Work by TSL and NBW was performed at University of Wisconsin - Madison and at Colorado State University for the
Jet Propulsion Laboratory, California Institute of Technology, sponsored by the National Aeronautics and Space Administration. The authors



**Table B1.** Backscatter and extinction properties for the snow particle model.

| $D_M$ | $C_{bk}$ | $C_{ext}$ | $D_M$ | $C_{bk}$ | $C_{ext}$ |
| mm | m$^2$ | m$^2$ | mm | m$^2$ | m$^2$ |
|---|---|---|---|---|---|
| 0.025 | 5.16253e-17 | 3.52024e-14 | 3.000 | 1.60708e-08 | 3.31323e-08 |
| 0.050 | 1.20475e-15 | 1.77890e-13 | 3.250 | 1.64139e-08 | 4.21088e-08 |
| 0.075 | 8.27182e-15 | 5.00664e-13 | 3.500 | 1.66119e-08 | 5.22600e-08 |
| 0.100 | 3.06847e-14 | 9.89618e-13 | 4.000 | 1.55864e-08 | 7.74061e-08 |
| 0.125 | 8.40498e-14 | 1.67138e-12 | 4.500 | 1.77554e-08 | 1.11609e-07 |
| 0.150 | 1.92314e-13 | 2.58903e-12 | 5.000 | 3.06798e-08 | 1.58860e-07 |
| 0.200 | 7.17250e-13 | 5.30976e-12 | 5.500 | 1.93705e-08 | 2.00667e-07 |
| 0.250 | 1.91396e-12 | 9.14038e-12 | 6.000 | 1.04092e-07 | 2.83080e-07 |
| 0.300 | 4.35436e-12 | 1.48681e-11 | 6.500 | 9.75512e-08 | 3.35067e-07 |
| 0.350 | 8.56280e-12 | 2.25304e-11 | 7.000 | 1.49787e-07 | 4.13047e-07 |
| 0.400 | 1.54761e-11 | 3.31046e-11 | 7.500 | 2.30734e-07 | 5.10512e-07 |
| 0.450 | 2.58963e-11 | 4.69409e-11 | 8.000 | 3.13418e-07 | 6.07034e-07 |
| 0.500 | 4.11161e-11 | 6.52071e-11 | 8.500 | 2.88081e-07 | 7.06213e-07 |
| 0.600 | 8.93929e-11 | 1.16523e-10 | 9.000 | 2.94080e-07 | 8.59293e-07 |
| 0.700 | 1.75650e-10 | 2.00793e-10 | 9.500 | 2.01596e-07 | 9.74101e-07 |
| 0.800 | 3.06043e-10 | 3.20526e-10 | 10.000 | 2.07686e-07 | 1.12076e-06 |
| 0.900 | 4.97542e-10 | 4.93081e-10 | 11.000 | 2.33291e-07 | 1.46061e-06 |
| 1.000 | 7.65231e-10 | 7.27755e-10 | 12.000 | 5.94999e-07 | 1.90676e-06 |
| 1.250 | 1.82454e-09 | 1.67311e-09 | 13.000 | 5.45403e-07 | 2.27318e-06 |
| 1.500 | 3.56830e-09 | 3.30096e-09 | 14.000 | 6.63279e-07 | 2.81244e-06 |
| 1.750 | 5.83188e-09 | 5.75812e-09 | 15.000 | 9.90939e-07 | 3.58772e-06 |
| 2.000 | 8.34684e-09 | 9.01675e-09 | 16.000 | 6.39329e-07 | 4.18269e-06 |
| 2.250 | 1.10293e-08 | 1.33794e-08 | 17.000 | 7.07551e-07 | 4.86569e-06 |
| 2.500 | 1.38623e-08 | 1.90102e-08 | 18.000 | 1.05353e-06 | 5.83543e-06 |
| 2.750 | 1.50482e-08 | 2.53761e-08 | | | |

also acknowledge Peter Rodriguez and David Hudak of Environment and Climate Change Canada for managing and making available C3VP observations used in this work.



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
