# Peer review of "What millimeter-wavelength radar reflectivity reveals about snowfall: An information-centric analysis"

_Atmospheric Measurement Techniques, 2020_

## Referee Comment (RC1) · Anonymous Referee #1 · 9 Aug 2020

This is a clearly written and carefully presented manuscript describing an optimal estimation retrieval of snowfall from W-band radar reflectivity and the associated uncertainties. In particular, the very neat description of the optimal estimation framework and of the separation of the different factors contributing to the uncertainty of the retrieval makes it a very good introductory paper on optimal estimation applied to snow retrieval. Overall, the research is sound, but I have two major concerns and few minor comments. I recommend the paper for publication after these points are addressed.

Major comments:

1) I think that the conclusions about the overall performances of the retrieval for the

whole C3VP field campaign is misleading and should be toned down. The retrieval presented in this paper requires assumptions on the snow particle properties that were obtained in Wood et al. (2015) by exploiting (at least partly) the same dataset. On one hand, I can accept that the observations used for deriving the properties of the snow particle "microphysical model" (mass-size and area-size parameters) can be considered as independent since the in-situ observations were combined with X-band radar measurements. On the other hand, the particle "scattering model" has been specifically selected to get the best match between W-band radar measurements and reflectivity computed from in-situ observations. It is therefore not surprising that the retrieval of the current paper provides an accumulated snowfall in such a good agreement with in-situ data. Therefore, I suspect that this impressive agreement is more due to a compensation of errors between the different snowfall cases than a very accurate performance of the retrieval as the instantaneous errors suggest. I have serious doubts that the same overall accuracy could be obtained when using a fully independent dataset. In the current version of the paper, a reader could really wonder why we would need more accurate observations of snow.

2) There is no question about the value of the optimal estimation retrieval described in this paper, in particular for assessing the different contributors to uncertainty. However, since the retrieval is applied to a single radar range gate and attenuation is neglected, and based on the conclusion that the W-band reflectivity is much more sensitive to log(lambda) than log(N0), it sounds that a much simpler retrieval (such as Z-lambda statistical relation) could be proposed with performances probably similar the optimal estimation proposed in this study. Please comment.

**Minor comments:**

1) L145: In order to emphasize that observations are independent, I would specify that X-band radar observations were used in Wood et al. (2015) for deriving the snow particle "microphysical model".

2) L300-303: Related to my major comment, by checking Wood et al. (2015), it appears that the 3 cases mentioned were indeed used for deriving the snow particle "microphysical model". However, practically the same overall dataset (Wood et al. (2015) say: "13 days from 2 December 2006 to 26 February 2007") was used to select the particle "scattering model".

3) Table 1: While I was trying to understand which part of the C3VP dataset was used for each part of the work in the current study and in Wood et al. (2015), I realized that the 3 common cases reported both in Wood et al. (2015) and in the current study don't show the same FD12P snow accumulation in LWE mm (0.8 vs 3.2 mm for the 6 Dec. 2006; 0.093 vs 10.2 mm for the 7 Dec. 2006 and 1.06 vs 4.6 mm for 27 Jan. 2007). These numbers are very different, please clarify. The accumulation is always smaller in the current paper. If the data on those days was only partially used, please explain why.

4) L381-388: It would help the understanding if you introduce the sensitivity of the forward model to both log(lambda) and log(N0) at the same time while saying that the sensitivity to N0 is not shown in a figure because it is constant and equal to 10.

Typos and awkward phrasing: 1) L57: evaluate is used twice 2) L125: "based on information theory" 3) L191: "Northwest" 4) L276: missing number after comma? 5) L444: reduce is used twice

СЗ

---

## Referee Comment (RC2) · Maximilian Maahn (Referee) · 19 Aug 2020

The authors present a snowfall retrieval based on radar reflectivity and temperature. While similar retrievals have been developed before, the focus of using only a single radar reflectivity and the extraordinary detailed error analysis makes it nevertheless an important contribution. The paper is well written, shows attention to detail, and the figures are clear. I have quite a few comments, but they are all of minor importance and I recommend the paper to be published subject to the following comments:

General comments:

[Figure]

- Does this paper describe how CloudSat's 2C-SNOW-PROFILE works? If yes, I would recommend to say so. If not, I would recommend to mention the differences

- I wonder how does this retrieval compares to traditional Ze-P relations? Clearly the sophisticated error estimates are an advantage, but what about the absolute P values? For several fixed temperature values, can the authors plot P as a function of Ze? This would allow to see 1) where the retrieval deviates from a power law form, 2) the impact of temperature on P, and 3) how it compares to published Ze-P relations

Specific comments

- L120: I assume the authors refer to the a posteriori covariance of x?

- L124: Is this the test shown in chapter 12.3.2 of Rodgers, 2000?

- L169: Do the authors underestimate Dmax when they use a measurement by an optical instrument? Isn't is quite unlikely that an individual particle is rotated such that the true Dmax can be observed?

- L174: does the log(N0) distribution have a Gaussian shape?

- L204: Do the authors think that their results are also applicable to high latitude locations?

- Figure 1) Why is the aircraft based N0 higher than the ground-based? Has the C3VP dataset been corrected for in situ probe shattering effects?

- L228: Typo in SF

- L233: I would say that Optimal Estimation cannot handle biases at all. I think it is perfectly acceptable that the authors assume that the CPR does not have any bias, but I would recommend to remove 'uncertainty in the absolute radiometric calibration'

- L247: Defining Kb is a very important step, I would recommend to spend 2-3 sentences on it instead of referring only to previous work.

- Figure 3: Add to the caption that measurement uncertainty is shown.

- L266: I appreciate that the authors do handle the errors sources conservatively and do not oversell the retrieval's uncertainty, but I wonder whether they are a little bit too pessimistic here: A radar always observes thousands of particles, isn't it quite unlikely that they are all of the same kind? Maybe a more recent bulk scattering method such as SSRGA would work better?

- L281: Why didn't the authors use the follow up paper by Heymsfield and Westbrook (2010)?

- L286: I would recommend to provide some details about S_b, I guess it contains the uncertainties of the m(D) relation?

- Figure 8: A more convincing evaluation example would be to use a different data set, e.g. from the high Arctic

- L345: I wonder whether the discussion about accumulation errors is relevant for CloudSat since it can provide only a snapshot of the current measurements?

- L376: Why is the number of states 0.9 higher than H?

- L378: A couple of years ago, I had the same problem and, after thinking about it a long time and checking my code many times, came to the same conclusion, i.e.

it is related to high correlations. However, I looked into the same issue recently and found that the negative values on the diagonal of A disappear after I added checks making sure that my covariance matrices are not singular: Python's (and I guess this applies to other languages, too) built-in inversion routine is quite forgiving and also inverts matrices that are 'slightly' singular. However, these instabilities can add up and many matrix inversions later lead to a negative entry on the diagonal of A. And it turned out I had created the singular matrix by myself by applying the authors' eq. 16 which added some numerical noise making my S_Epsilon singular and non-symmetric. After making sure that my S_Epsilon is really symmetric and nonsingular (i.e. doing a rank test), negative values on the diagonal of A disappeared. I admit I never investigated this systematically, so it could be a coincidence, but I would be curious to see whether the authors' negative values are also related to numerical instabilities. In the end, this appears to be a cosmetic issue and not very important: the total degrees of freedom and all other results stayed the same in my sample retrieval.

• Figure 12: I'm surprised that the d_s values are not higher. A couple years ago I developed an ice cloud retrieval where I, because it was an information content study, simply added everything to the state vector: 3 PSD parameters, 2 m(D) parameters, and 2 A(D) parameters. With such a large state vector, I always got two d_s when using Ze and mean Doppler (I never tried only Ze). The fact that d_s is not 1 in the authors' study might mean that a little bit of information is unused. I wonder whether this information could be used and d_s would be 1 if the m(D) parameters were moved from the b vector to the x vector. This might lead to lower P uncertainties. This would also help with the issue raised in L419. Of course, this includes the challenge to make sure that the retrieval doesn't put all the information into m(D) instead of the PSD parameters which probably would make the P uncertainty even larger.

• L450: Add the DOI

Heymsfield, A. J., and C. D. Westbrook, 2010: Advances in the Estimation of Ice Particle Fall Speeds Using Laboratory and Field Measurements. J. Atmos. Sci., 67, 2469–2482, doi:10.1175/2010JAS3379.1.

---

## Referee Comment (RC3) · Anonymous Referee #3 · 21 Aug 2020

This manuscript describes the optimal retrieval process for deriving snowfall from single millimeter-wavelength radar reflectivity observations and represents a derived uncertainty analysis divided into the relevant error sources in the process. The manuscript is logically and well structured, and fluently written. The topic has been walked through by describing the steps nicely and in an informative way, and the manuscript was very nice to read. Although the topic is not necessarily very novel, when more recently the tendency has been on describing the multi-frequency retrieval processes or retrievals combining single-frequency with Doppler-velocity information, the way the authors describe the details in the process, and especially, the factors influencing the uncertainty, the manuscript has a significant contribution and is relevant and interesting for the

community. My recommendation is that the manuscript should be published with minor changes and clarifications.

Below I have a few small requests for clarifications for the authors.

Request for clarification:

1. My major concern in the development of the retrieval process is that it is, if I have understood correctly, based on almost single simplified snow model (Wood et al.2015) with single values of $\alpha$, $\beta$, $\gamma$ and $\delta$ with a certain uncertainty stated in Appendix B. Although, it has been shown e.g. in Kulie and Bennartz, 2009 in Figure 1, the huge variety in backscattering coefficients for different particle types and as it is stated also in the manuscript e.g. on line 246 that in Wood et al. 2015 the uncertainties for the described perturbed particle models can be as high as 15 dB. What about the effect of aggregation creating very different particle properties (low area ratio) from single crystals or rimed particles? I wonder could this be the reason for the two outlier cases described in the results in paragraph starting on line 319 that the particle types (m(D) and v(D) were so different from the used parametrization of Mitchel and Heymsfield 2005), and the used particle model were not descriptive for the observed particles, al-though still dry snow particles and not e.g. melting as proposed in the manuscript on line 403. And therefore, the retrieval failed. Actually, looking at the Figure 7 with all higher snowfall rates (> 0.8 mm), the retrieval seems to underestimate quite systemat-ically. If seen also as relevant, could the authors provide more discussion on this topic to the manuscript although there is already the statement on line 403?

2. As a second point, in my understanding, the Z-S retrieval is less dependent on the intercept parameter in the millimeter wavelength than in the centimeter region where the Rayleigh approximation is applicable (e.g. Rasmussen et al. 2003, Bukovčić et al. 2018). In the abstract, it is stated on line 14, that the PSD intercept is less well constrained by the retrieval, and in Appendix on line 488, that the measurements better sensitive to log(N0) could benefit the retrieval. Could the authors elaborate, whether

these results refer that the dependence on N0 is physically no longer similar significant to the retrieval.

And continuing, would then actually simple averaged Z-S relations provide equally good retrieval results, in case the particle type and shape are correctly modeled? The retrieval results here are compared to the collocated measurements of accumulation with a weather sensor. It would be also interesting to see, how different are the snowfall rates with this presented optimal retrieval method in comparison to these simple Z-S relations presented in the literature such as e.g. Kulie and Bennartz, 2009 or Matrosov, 2007.

Small comments:

In the paragraph starting on line 29, the ground-based radar observations are described. It is slightly striking that only MRR related literature references are stated, even though this paper is describing the retrieval process utilized for the W-band. Was there a certain reason for this choice, why e.g. other studies presenting W-band observations were left out? Although, some of the radars are mentioned on lines 71-73.

Figure 11. Suggestion to use a different color for FD12P to improve readability.

Line 381 Suggestion to use a different term for shape of the size distributions. Although it is clear that the exponential PSD distribution is used here and this term describes the effect of lambda in the metric curves in Figure 13, it still is close to the widely used shape parameter $\mu$ of gamma PSD.

Typos:

Line 307 should be Vaisala FD12P (Vaisala Oyj, 2002) instead of (Viasala Oyj, 2002) Line 336 remove the unit

Rasmussen, R., M. Dixon, S. Vasiloff, F. Hage, S. Knight, J. Vivekanandan, and M. Xu, 2003: Snow Nowcasting Using a Real-Time Correlation of Radar Reflectivity with Snow Gauge Accumulation. J. Appl. Meteor., 42, 20–36, https://doi.org/10.1175/1520-

0450(2003)042<0020:SNUART>2.0.CO;2.

Bukovčić, P., A. Ryzhkov, D. Zrnić, and G. Zhang, 2018: Polarimetric Radar Relations for Quantification of Snow Based on Disdrometer Data. J. Appl. Meteor. Climatol., 57, 103–120, https://doi.org/10.1175/JAMC-D-17-0090.1.

Kulie, M. S., and R. Bennartz, 2009: Utilizing spaceborne radars to retrieve dry snowfall. Journal of Applied Meteorology and Climatology,48 (12), 2564–2580, doi:10.1175/2009JAMC2193.1.

Matrosov, S. Y., 2007: Modeling backscatter properties of snowfall at millimeter wavelengths.619Journal of the Atmospheric Sciences,64 (5), 1727–1736, doi:10.1175/JAS3904.1.

---

## Author Comment (AC2) · 24 Sep 2020

Dear Dr. Maahn,

Thank you very much, we appreciate your comments and suggestions. Our responses to your general and specific comments are as follows:

Responses to general comments:

1. Yes, with the exception that the retrieval here operates on single-range-bin reflectivity observations (and for this analysis involves no treatments for attenuation, multiple scattering or spatial correlation), this is the retrieval method used for CloudSat's 2C-

[Figure]

SNOW-PROFILE. A companion paper is near completion - it extends this retrieval to CloudSat Cloud Profiling Radar observations. We have added a statement indicating the relationship to the CloudSat retrieval in the introduction at line 74.

2. Yes, we've prepared a figure that shows the temperature dependence of the Z-S results obtained for this retrieval and makes comparisons of our results against a number of published Z-S relationships. It is included as Figure 7 in section 4 of the revised manuscript. Our results are consistent with several of these relationships at reflectivities up to about 5-7 dBZe. Above 7 dBZe, our results tend toward smaller snowfall values than a purely linear relationship would produce. The smaller values are more consistent with the Kulie and Bennartz (2009) HA particle results, which represent an aggregate particle. For our results, snowfall rates at a given Z become larger generally with warming temperature.

Responses to specific comments:

L120: Yes, it is the current, iterative estimate of the a posteriori covariance of x, exactly. We have reworded the description to provide this information.

L120: Yes, This test is actually obtained from Marks and Rodgers (1993) from their equation (16) and the discussion that follows, but it is the implementation of the test for correct convergence described by Rodgers (2000) in section 12.3.2. We now also include a citation of Marks and Rodgers.

L169: Yes, the optical instruments do underestimate Dmax (see Wood et al., 2013, for example). The retrieval used in Wood et al. (2015) to determine the microphysical and scattering a priori properties used in this retrieval includes compensation for this effect. Accordingly, the a priori properties used in this retrieval are based on Dmax.

L174: Yes, much more so that would N0 itself. Please see the figure we've provided.

L204: Yes, although that opinion is based on tests done with the actual CloudSat retrieval product in comparisons against ground-based observations (primarily in Antarctica and Sweden).

Figure 1: The differences appear more substantial than what would be attributable to shattering (this is based on a quick look at the corrected versus uncorrected distributions in Field et al., 2006, JTECH). The aircraft data in Figure 1 include observations well above the surface. We expect that the differences are largely due to microphysical processing between locations aloft and the surface. Documentation for the C3VP aircraft 2D probe particle data do not indicate whether shattered particle correction was performed.

L228: Corrected, thank you.

L233: We agree, but we've revised this sentence somewhat differently than suggested. We indicate that both bias and measurement noise contribute to reflectivity errors, but that we've used the CPR noise characteristics to estimate $S_y$.

L247: This paragraph was revised to provide more details about $K_b$.

Figure 3: Done, the caption was changed to indicate that measurement uncertainty is shown.

L266: Another way to look at this question is to ask, given two different radar volumes filled with particles that follow exactly the same m(D), same Ap(D) and same size distribution but whose particle shapes are not constrained to match each other, does it seems reasonable that their reflectivities could differ by about 2 dB? That seems possible, but yes, may be conservative especially given that, for snowfall, we are often observing populations of irregular aggregates rather than pristine particles..

L281: There was no strong reason for not using Heysfield and Westbrook (2010) for fallspeeds. As part of other work (that is described in Wood et al., 2014, 2015), we performed tests by switching between HW2010 and Mitchell and Heymsfield (2005) and found little impact on those retrieval results, but we should revisit our choice for this retrieval.

L286: We have revised equation 19 slightly to clarify that S_{\tilde{b}} is the uncertainty in snowfall rate that results from the uncertainties in the particle model parameters, and added a description of how it is calculated.

Figure 8: We agree there would be value in comparing against other datasets, but this comparison does illustrate the effects of substantial departures from the a priori assumptions of the retrieval and the behavior of accumulation errors. We do plan to apply this retrieval method to other field experiment datasets that involve ground-based radars.

L345: We are examining the calculation of accumulations from intermittent observations (such as provided by CloudSat) and the resulting errors for the companion paper. We agree, it isn't clear that the treatment here for a fixed radar taking essentially continuous meausrements would be highly applicable to CloudSat.

L376: It's a bit of a numerical coincidence due to the particular values of H. The number of states is given by 2**H (described well in the L'Ecuyer et al., 2006 reference).

L378: Thanks very much for providing this information. In this single-bin reflectivity-only retrieval, the S_Epsilon matrix consists of a single element, so is never ill-conditioned. The other matrix that must be inverted and that is used in the A-matrix calculation is the a priori covariance, S_a; its condition number is around 20. The final matrix that must be inverted has condition numbers ranging from 35 to 95. These are somewhat large, but indicate a potential loss of precision of only 1-2 decimal places in the inverse calculation (which uses double-precision arithmetic). Based on these values, we think it seems less likely that ill-conditioning is the source of the negative A-matrix values.

Figure 12: This is an interesting interpretation. Yes, it seems it could be an indication that the retrieval is prevented from fully utilizing the information in the reflectivity observation. It would be interesting to try including m(D) parameters in the state vector, maybe using a Rayleigh regime reflectivity forward model, just as a test.

[Figure]

L450: Unfortunately, the effort by the GHRC to archive the C3VP observations (including those used in this study) was initiated as a result of this manuscript and is not complete. We are looking at alternate archive locations and will need to add the necessary information prior to the manuscript being accepted by AMT.
* * *
[Figure]

[Figure]

**Fig. 1.** Histograms of N0 and log(N0)

[Figure]

---

## Author Comment (AC3) · 24 Sep 2020

Dear Reviewer #3, thank you for your feedback and requests for clarification. Please see our explanations below of our responses to your comments.

Requests for clarification

1. Yes, aggregation could lead to dry snow particles with properties that are very different from those used as a priori assumptions in the retrieval. In general, aggregation (and other microphysical processes) may change how mass, area, fallspeed, and scattering properties all vary with particle size D. The retrieval has some freedom to adapt

to this by altering the retrieved log(lambda) and log(N0) so that the forward-modeled Ze matches the observed Ze. Large differences, however, will likely lead to nontrivial retrieval errors.

For the 14 February anomaly, in addition to the evidence mentioned briefly in the manuscript, a collocated X-band Doppler radar (McGill University's VertiX) revealed a bright band at around 1 km AGL with Doppler velocities of around 3 m/s below this level. This is a large fallspeed for dry snow aggregates, but these might have been large, wet aggregates. The VertiX was not in operation for the 2 March anomaly. It's probably not possible to rule out that aggregation was involved in either anomaly.

Yes, there does seem to be underestimation for higher snowfall rates in Figure 7, balanced by overestimation to some degree as evidenced by events with positive fractional differences in accumulations shown in Table 1. This suggest there might be a benefit for making the particle model a function of the observed reflectivity, and should be investigated further.

We have revised the statement in the Discussion section (at about line 425 in the revised manuscript) to be more consistent with this clarification.

We have also added brief commentary in the 4th paragraph of section 4 (around line 340 of the revised manuscript).

2. Yes, log(N0) is still physically significant to the retrieval, since it does influence Ze and snowfall rate; however log(lambda) is more strongly constrained by the reflectivity measurement than is log(N0). That means that the retrieval must employ either "a priori" information or additional measurements to help constrain log(N0).

Regarding simple, averaged Z-S relationships, please see Figure 7 that has been added to the manuscript. The retrieval gives Z-S relationships that are temperature-dependent and that deviate from the plotted linear relationships obtained from published Z-S relationships. It seems likely that temperature-dependent functions could be

developed that mimic the Z-S behavior of the retrieval (but not the information-centric diagnostics). Please see also our response to reviewer #1.

Small comments:

Paragraph starting on line 29: No, there wasn't a conscious decision to omit W-band radars. For experiments like these, W-band radars, because of their primary application as cloud radars, are more often operated for cloud measurement from aircraft and less often for observing snowfall at the ground. We know that GCPEx and ICE-POP in addition to C3VP all deployed ground-based W-band radars. Two Department of Energy funded experiments (StormVEx, at Colorado's Storm Peak Laboratory and BAECC in Finland), also deployed ground-based W-band radars. A number of studies have used W-band radar observations as part of multi-frequency snowfall retrievals, which are not applicable to this work. Development of snowfall retrievals using single-frequency, ground-based radar observations at W-band is not common. Some work has been done using ground-based Ka-band radars for snowfall (e g., Matrosov et al., 2008, JAMC; Cooper et al., 2017)

We have revised the text starting around line 35 in the revised manuscript to include information about experiments with ground-based W-band radars.

Figure 11: Done, and symbol sizes were increased to improve clarity.

Line 381: We changed this to say simply that "The size distribution plays a significant role..."

Typos:

1. "Viasala Oyj" is changed to "Vaisala Oyj".

2. Regarding "unit" appearing on line 336, corrected, thanks.

Matrosov, S.Y., M.D. Shupe, and I.V. Djalalova, 2008: Snowfall retrievals using millimeter-wavelength cloud radars. J. Appl. Meteor. Climatol., 46, 769-777.

doi:10.1175/2007JAMC1768.1

Cooper S. J., N. B. Wood, and T. S. L'Ecuyer, 2017: A variational technique to estimate snowfall rate from coincident radar, snowflake, and fallspeed observations. Atmos. Meas. Tech., 10, 2557-2571, doi:10.5194/amt-10-2557-2017.

———————————————————

---

## Author Response (AR1)

Response to Reviewer #1
######################

Dear Reviewer #1,

Thanks very much for your comments and suggestions, which we have generally adopted. Our responses to the major and minor comments follow below with your original comments marked by leading ">" characters. In our responses, line numbers denoted as "Lxxx" refer to the original document and "DLxxx" refer to the marked-up version of the new document.

> This is a clearly written and carefully presented manuscript describing an
> optimal estimation retrieval of snowfall from W-band radar reflectivity and
> the associated uncertainties. In particular, the very neat description of the
> optimal estimation framework and of the separation of the different factors
> contributing to the uncertainty of the retrieval makes it a very good
> introductory paper on optimal estimation applied to snow retrieval. Overall,
> the research is sound, but I have two major concerns and few minor comments. I
> recommend the paper for publication after these points are addressed.

> Major comments:

> 1) I think that the conclusions about the overall performances of the
> retrieval for the whole C3VP field campaign is misleading and should be toned
> down. The retrieval presented in this paper requires assumptions on the snow
> particle properties that were obtained in Wood et al. (2015) by exploiting (at
> least partly) the same dataset. On one hand, I can accept that the
> observations used for deriving the properties of the snow particle
> "microphysical model" (mass-size and area-size parameters) can be considered
> as independent since the in-situ observations were combined with X-band radar
> measurements. On the other hand, the particle "scattering model" has been
> specifically selected to get the best match between W-band radar measurements
> and reflectivity computed from in-situ observations. It is therefore not
> surprising that the retrieval of the current paper provides an accumulated
> snowfall in such a good agreement with in-situ data. Therefore, I suspect that
> this impressive agreement is more due to a compensation of errors between the
> different snowfall cases than a very accurate performance of the retrieval as
> the instantaneous errors suggest. I have serious doubts that the same overall
> accuracy could be obtained when using a fully independent dataset. In the
> current version of the paper, a reader could really wonder why we would need
> more accurate observations of snow.

Thanks for this feedback. It seems likely that in this comment "overall performances" refers principally to our description of the agreement of retrieved and observed precipitation rates (e. g., Figure 8 of the revised document) and of the agreement in seasonal accumulation achieved by the retrieval in comparison to that obtained from measured values. We have extended the statement at the end of Section 4, paragraph 1 to elaborate on the C3VP data's roles in the development of the microphysical and scattering models. Further, we have added a statement near the beginning of Section 4, paragraph 4 indicating that the agreement of observed and retrieved snowfall rates is not unexpected. A similar statement regarding the accumulation comparisons has been added near the beginning of section 4, paragraph 5. Finally, we reiterated this point in paragraph 2 of section 5 (Discussion and conclusions). See DL320, DL337, DL350, DL437.

Other principal results discussed in section 5 include the instantaneous retrieval uncertainties and sources of uncertainties in the retrieved state (paragraph 3), information content (paragraph 4), sources of model-measurement uncetainties (paragraph 5). These results depend mostly on the estimates of

observation and forward model uncertainties plus forward model sensititivies and would be at most only weakly sensitive to the concerns raised by the reviewer.

We would appreciate further feedback if these modifications do not target the particular issue intended by the reviewer.

> 2) There is no question about the value of the optimal estimation retrieval
> described in this paper, in particular for assessing the different
> contributors to uncertainty. However, since the retrieval is applied to a
> single radar range gate and attenuation is neglected, and based on the
> conclusion that the W-band reflectivity is much more sensitive to log(lambda)
> than log(N0), it sounds that a much simpler retrieval (such as Z-lambda
> statistical relation) could be proposed with performances probably similar the
> optimal estimation proposed in this study. Please comment.

One of the benefits of this retrieval approach is that the retrieval can, on-the-fly, determine appropriate weights to apply to the information in the observations versus information provided by the a priori. Since Z alone cannot uniqely determine S, a priori information of some form is required. For simple statistical retrievals, this a priori information is generally embedded in the statistical relationships. It's not clear to us that this information-based weighting would be provided in a simple statistical retrieval.

That said, simple temperature-dependent statistical relationships that would provide estimates of snowfall rate and their uncertainties could be constructed. See, for example, Figure 7 of the revised manuscript to see how Z-S for this retrieval varies with temperature. There are some drawbacks:

First the differences in sensitivity and information that are made explicit with this method tell us quantitatively that a Z-lambda approach would not be sufficient. It's clear that the retrieval requires information about N0 that is not provided well by the Z measurement, so a priori information about N0 is required. Diagnostics like this would not be obtainable in a simple statistical retrieval.

Second, consider what must be done when observed snowfall rate values are found to depart substantially from the retrieved values (i.e., the retrieval fails). With this method it's straightforward to compare the retrieval's assumptions, which are explicit, against observations to determine the cause of the retrieval failure. With a statistical approach, in which the a priori assumptions are typically not explicit, the causes of retrieval failure are much less transparent.

> Minor comments:

> 1) L145: In order to emphasize that observations are independent, I would
> specify that X-band radar observations were used in Wood et al. (2015) for
> deriving the snow particle "microphysical model".

1) Done. We have revised the referenced sentence (in the first paragraph of section 2.1.1, in the text following equation 8) to read "That work used in-situ measurements and remotely-sensed X-band reflectivity observation of snow from C3VP...". (DL151)

> 2) L300-303: Related to my major comment, by checking Wood et al. (2015), it
> appears that the 3 cases mentioned were indeed used for deriving the snow
> particle "microphysical model". However, practically the same overall dataset
> (Wood et al. (2015) say: "13 days from 2 December 2006 to 26 February 2007")

> was used to select the particle "scattering model".

2) Yes, this is correct. The microphysical properties (m(D) and A(D)) and generic shape (which with m(D) and A(D) determine the scattering properties) used different ranges of the C3VP observations owing to differences in the availability of the required observations.

In W15, first, observations from 4 snowfall events were used to estimate the PDFs of microphysical properties (m(D) and A(D) but not shape). The data described in Table 1 of this manuscript for 6-7 December 2006 and 26-27 January 2007 are from small time periods of three of these events (SYN1, LE1, and LE2) during which the ACR was operated. Second, given m(D) and A(D), particles of different generic shapes or habits were modeled and radar scattering properties were calculated. ACR observations from the "13 days from 2 December 2006 to 26 February 2007" were then used to determine the generic shape that best reproduced the observed reflectivities. See our response to major comment #1.

> 3) Table 1: While I was trying to understand which part of the C3VP dataset
> was used for each part of the work in the current study and in Wood et al.
> (2015), I realized that the 3 common cases reported both in Wood et al. (2015)
> and in the current study don't show the same FD12P snow accumulation in LWE mm
> (0.8 vs 3.2 mm for the 6 Dec. 2006 ; 0.093 vs 10.2 mm for the 7 Dec. 2006 and
> 1.06 vs 4.6 mm for 27 Jan. 2007).  These numbers are very different, please
> clarify. The accumulation is always smaller in the current paper. If the data
> on those days was only partially used, please explain why.

3.  While the FD12P and other instruments involved in the study were mostly autonomous and ran continuously during snowfall events, the ACR required an attending operator and so ran for shorter periods of time within the events. Note that the durations shown in Table 1 are substantially shorter than those shown for the events used in W15.

> 4) L381-388: It would help the understanding if you introduce the sensitivity
> of the forward model to both log(lambda) and log(N0) at the same time while
> saying that the sensitivity to N0 is not shown in a figure because it is
> constant and equal to 10.

4.  We have revised the paragraph (3rd paragraph of section 4.2, near line 405 of the revised manuscript) to introduce earlier the constant sensitivity to log(N0) in contrast to the varying sensitivity to log(lambda). (DL412)
Typos and awkward phrasing:

1) L57: evaluate is used twice

The second instance of "evaluating" has been changed to "estimating".

2) L125: "based on information theory"

"based in" is our intended wording.

3) L191: "Northwest"

"Northewest" has been corrected.  (DL197)

4) L276: missing number after comma?

To clarify, we have written this as "0.00", which is the actual value to two decimal places.  (DL287)

> 5) L444: reduce is used twice

"reduced by reducing" is our intended wording.

Response to Reviewer #2
######################

Dear Dr. Maahn,

Thank you very much, we appreciate your comments and suggestions. Our
responses to your general and specific comments follow with your original
comments marked with leading ">" characters. In our responses, line numbers
denoted as "Lxxx" refer to the original document and "DLxxx" refer to the
marked-up version of the new document.

> The authors present a snowfall retrieval based on radar reflectivity and
> temperature. While similar retrievals have been developed before, the focus
> of using only a single radar reflectivity and the extraordinary detailed error
> analysis makes it nevertheless an important contribution. The paper is well
> written, shows attention to detail, and the figures are clear. I have quite a
> few comments, but they are all of minor importance and I recommend the paper
> to be published subject to the following comments:
>
> Does this paper describe how CloudSat's 2C-SNOW-PROFILE works? If yes, I
> would recommend to say so. If not, I would recommend to mention the
> differences

Yes, with the exception that the retrieval here operates on
single-range-bin reflectivity observations (and for this analysis involves no
treatments for attenuation, multiple scattering or spatial correlation), this
is the retrieval method used for CloudSat's 2C-SNOW-PROFILE. A companion
paper is near completion - it extends this retrieval to CloudSat Cloud
Profiling Radar observations. We have added a statement indicating the
relationship to the CloudSat retrieval in the introduction. (DL76)

> I wonder how does this retrieval compares to traditional Ze-P relations?
> Clearly the sophisticated error estimates are an advantage, but what about the
> absolute P values? For several fixed temperature values, can the authors plot
> P as a function of Ze? This would allow to see 1) where the retrieval deviates
> from a power law form, 2) the impact of temperature on P, and 3) how it
> compares to published Ze-P relations

Yes, we've prepared a figure that shows the temperature dependence of the
Z-S results obtained for this retrieval and makes comparisons of our results
against a number of published Z-S relationships. It is included as Figure 7
in section 4 of the revised manuscript. Our results are consistent with
several of these relationships at reflectivities up to about 5-7 dBZe. Above
7 dBZe, our results tend toward smaller snowfall values than a purely linear
relationship would produce. The smaller values are more consistent with the
Kulie and Bennartz (2009) HA particle results, which represent an aggregate
particle. For our results, snowfall rates at a given Z become larger
generally with warming temperature.

> Specific comments

> L120: I assume the authors refer to the a posteriori covariance of x?

Yes, it is the current, iterative estimate of the a posteriori
covariance of x, exactly. We have reworded the description to provide
this information. (DL124)

> L124: Is this the test shown in chapter 12.3.2 of Rodgers, 2000?

Yes, This test is actually obtained from Marks and Rodgers (1993) from their
equation (16) and the discussion that follows, but it is the implementation of
the test for correct convergence described by Rodgers (2000) in section

12.3.2.  We now also include a citation of Marks and Rodgers. (DL129)

> L169: Do the authors underestimate Dmax when they use a measurement by an
> optical instrument? Isn't is quite unlikely that an individual particle is
> rotated such that the true Dmax can be observed?

The optical instruments do underestimate Dmax (see Wood et al., 2013,
for example).  The retrieval used in Wood et al. (2015) to determine the
microphysical and scattering a priori properties used in this retrieval
includes compensation for this effect.  Accordingly, the particle a priori
properties used in this retrieval are based on Dmax and so the retrieved size
distribution parameters are those for Dmax.

> L174: does the log(N0) distribution have a Gaussian shape?

Yes, much more so that would N0 itself.  Please see the figure
N0_Dmax_histograms_SVI_C3VP.png we provided with our online discussion
comments.

> L204: Do the authors think that their results are also applicable to high
> latitude locations?

Yes, although that opinion is based on tests done with the actual
CloudSat retrieval product in comparisons against ground-based observations
(primarily in Antarctica and Sweden).  Please see for example Lemmonier et
al., 2020, doi:10.1029/2019JD031399.

> Figure 1) Why is the aircraft based N0 higher than the ground-based? Has the
> C3VP dataset been corrected for in situ probe shattering effects?

The differences appear more substantial than what would be
attributable to shattering (this is based on a quick look at the corrected
versus uncorrected distributions in Field et al., 2006, JTECH).  The aircraft
data in Figure 1 include observations well above the surface.  We expect that
the differences are largely due to microphysical processing between locations
aloft and the surface.  Unfortunately, documentation for the C3VP aircraft 2D
probe particle data do not indicate whether shattered particle correction was
performed.

> L228: Typo in SF

Corrected, thank you. (DL234)

> L233: I would say that Optimal Estimation cannot handle biases at all. I
> think it is perfectly acceptable that the authors assume that the CPR does not
> have any bias, but I would recommend to remove 'uncertainty in the absolute
> radiometric calibration'

We agree, but we've revised this sentence somewhat differently than
suggested.  We indicate that both bias and measurement noise contribute to
reflectivity errors, but that we've used the CPR noise characteristics to
estimate $S_y$.  (DL239, 241)

> L247: Defining Kb is a very important step, I would recommend to spend 2-3
> sentences on it instead of referring only to previous work.

This paragraph was revised to provide more details about $K_b$. (DL252)

> Figure 3: Add to the caption that measurement uncertainty is shown.

Done, the caption was changed to indicate that measurement
uncertainty is shown.  (Near DL256)

> L266: I appreciate that the authors do handle the errors sources
> conservatively and do not oversell the retrieval's uncertainty, but I wonder
> whether they are a little bit too pessimistic here: A radar always observes
> thousands of particles, isn't it quite unlikely that they are all of the same
> kind? Maybe a more recent bulk scattering method such as SSRGA would work
> better?

Another way to look at this question is to ask, given two different
radar volumes filled with particles that follow exactly the same m(D), same
Ap(D) and same size distribution but whose particle shapes are not constrained
to match each other, does it seems reasonable that their reflectivities
could differ by about 2 dB?  That seems possible, but yes, may be
conservative especially given that, for snowfall, we are often observing
populations of irregular aggregates rather than pristine particles..

> L281: Why didn't the authors use the follow up paper by Heymsfield and
> Westbrook (2010)?

There was no strong reason for not using Heysfield and Westbrook (2010)
for fallspeeds.  As part of other work (described in Wood et al., 2014,
2015), we performed tests by switching between HW2010 and Mitchell and
Heymsfield (2005) and found little impact on those retrieval results, but we
should revisit our choice for this retrieval.

> L286: I would recommend to provide some details about S_b, I guess it
> contains the uncertainties of the m(D) relation?

We have revised equation 19 slightly to clarify that S_{\tilde{b}}
is the uncertainty in snowfall rate that results from the uncertainties in the
particle model parameters, and added a description of how it is calculated.
(DL301)

> Figure 8: A more convincing evaluation example would be to use a different
> data set, e.g. from the high Arctic

We agree there would be value in comparing against other datasets,
but this comparison does illustrate the effects of substantial departures from
the a priori assumptions of the retrieval and the behavior of accumulation
errors.  We do plan to apply this retrieval method to other field experiment
datasets that involve ground-based radars.

> L345: I wonder whether the discussion about accumulation errors is relevant
> for CloudSat since it can provide only a snapshot of the current measurements?

We are examining the calculation of accumulations from intermittent
observations (such as provided by CloudSat) and the resulting errors for the
companion paper.  We agree, it isn't clear that the treatment here for
a fixed radar taking essentially continuous meausrements would be highly
applicable to CloudSat.

> L376: Why is the number of states 0.9 higher than H?

It's a bit of a numerical coincidence due to the particular values of
H.  The number of states is given by 2**H (described well in the L'Ecuyer et
al., 2006 reference).

> L378: A couple of years ago, I had the same problem and, after thinking
> about it a long time and checking my code many times, came to the same
> conclusion, i.e. it is related to high correlations. However, I looked into
> the same issue recently and found that the negative values on the diagonal of
> A disappear after I added checks making sure that my covariance matrices are
> not singular: Python's (and I guess this applies to other languages, too)
> built-in inversion routine is quite forgiving and also inverts matrices that
> are 'slightly' singular. However, these instabilities can add up and many
> matrix inversions later lead to a negative entry on the diagonal of A. And it
> turned out I had created the singular matrix by myself by applying the
> authors' eq. 16 which added some numerical noise making my S_Epsilon singular
> and non-symmetric. After making sure that my S_Epsilon is really symmetric and
> nonsingular (i.e. doing a rank test), negative values on the diagonal of A
> disappeared. I admit I never investigated this systematically, so it could be
> a coincidence, but I would be curious to see whether the authors' negative
> values are also related to numerical instabilities. In the end, this appears
> to be a cosmetic issue and not very important: the total degrees of freedom
> and all other results stayed the same in my sample retrieval.

Thanks very much for providing this information. In this single-bin
reflectivity-only retrieval, the S_Epsilon matrix consists of a single
element, so is never ill-conditioned. The other matrix that must be inverted
and that is used in the A-matrix calculation is the a priori covariance, S_a;
its condition number is around 20. The final matrix that must be inverted has
condition numbers ranging from 35 to 95. These are somewhat large, but
indicate a potential loss of precision of only 1-2 decimal places in the
inverse calculation (which uses double-precision arithmetic). Based on these
values, we think it seems less likely that ill-conditioning is the source of
the negative A-matrix values.

> Figure 12: I'm surprised that the d_s values are not higher. A couple years
> ago I developed an ice cloud retrieval where I, because it was an information
> content study, simply added everything to the state vector: 3 PSD parameters,
> 2 m(D) parameters, and 2 A(D) parameters. With such a large state vector, I
> always got two d_s when using Ze and mean Doppler (I never tried only Ze). The
> fact that d_s is not 1 in the authors' study might mean that a little bit of
> information is unused. I wonder whether this information could be used and d_s
> would be 1 if the m(D) parameters were moved from the b vector to the x
> vector. This might lead to lower P uncertainties. This would also help with
> the issue raised in L419. Of course, this includes the challenge to make sure
> that the retrieval doesn't put all the information into m(D) instead of the
> PSD parameters which probably would make the P uncertainty even larger.

This is an interesting interpretation. Yes, it seems it could be
an indication that the retrieval is prevented from fully utilizing the
information in the reflectivity observation. It would be interesting to try
including m(D) parameters in the state vector. This would require the forward
model to update particle scattering properties as m(D) parameters are
adjusted, so a less computationally-intensive method (compared to DDA) would
need to be used to calculate those properties - perhaps the SSRGA approach you
mentioned earlier.

> L450: Add the DOI

At this time, the effort by the GHRC to archive the C3VP observations
(including those used in this study) was initiated as a result of this manuscript
and is not complete. We are looking at alternate archive locations and will
need to add the necessary information prior to final publication. We have
added a placeholder data availability statement. (DL535)

> Heymsfield, A. J., and C. D. Westbrook, 2010: Advances in the Estimation of
> Ice Particle Fall Speeds Using Laboratory and Field Measurements. J. Atmos.
> Sci., 67, 2469-2482, doi:10.1175/2010JAS3379.1.

[Figure]

Figure 1: N0_Dmax_histograms_SVI_C3VP.png

Response to Reviewer #3
#######################

Dear Reviewer #3,

Thank you for your feedback and requests for clarification. Our responses to your comments follow below with your original comments marked with leading ">" characters. in our responses, line numbers denoted as "Lxxx" refer to the original document and "DLxxx" refer to the marked-up version of the new document.

> This manuscript describes the optimal retrieval process for deriving snowfall
> from single millimeter-wavelength radar reflectivity observations and
> represents a derived uncertainty analysis divided into the relevant error
> sources in the process. The manuscript is logically and well structured, and
> fluently written. The topic has been walked through by describing the steps
> nicely and in an informative way, and the manuscript was very nice to read.
> Although the topic is not necessarily very novel, when more recently the
> tendency has been on describing the multi-frequency retrieval processes or
> retrievals combining single-frequency with Doppler-velocity information, the
> way the authors describe the details in the process, and especially, the
> factors influencing the uncertainty, the manuscript has a significant
> contribution and is relevant and interesting for the community. My
> recommendation is that the manuscript should be published with minor changes
> and clarifications.

> Below I have a few small requests for clarifications for the authors.

> Request for clarification:

> 1. My major concern in the development of the retrieval process is that it is,
> if I have understood correctly, based on almost single simplified snow model
> (Wood et al.2015) with single values of (alpha, beta, gamma, sigma) with a
> certain uncertainty stated in Appendix B. Although, it has been shown e.g. in
> Kulie and Bennartz, 2009 in Figure 1, the huge variety in backscattering
> coefficients for different particle types and as it is stated also in the
> manuscript e.g. on line 246 that in Wood et al. 2015 the uncertainties for the
> described perturbed particle models can be as high as 15 dB. What about the
> effect of aggregation creating very different particle properties (low area
> ratio) from single crystals or rimed particles? I wonder could this be the
> reason for the two outlier cases described in the results in paragraph
> starting on line 319 that the particle types (m(D) and v(D) were so different
> from the used parametrization of Mitchel and Heymsfield 2005), and the used
> particle model were not descriptive for the observed particles, although still
> dry snow particles and not e.g. melting as proposed in the manuscript on line
> 403. And therefore, the retrieval failed. Actually, looking at the Figure 7
> with all higher snowfall rates (> 0.8 mm), the retrieval seems to
> underestimate quite systematically. If seen also as relevant, could the
> authors provide more discussion on this topic to the manuscript although there
> is already the statement on line 403?

Yes, aggregation could lead to dry snow particles with properties that are very different from those used as a priori assumptions in the retrieval. In general, aggregation (and other microphysical processes) may change how mass, area, fallspeed, and scattering properties all vary with particle size D. The retrieval has some freedom to adapt to this by altering the retrieved log(lambda) and log(N0) so that the forward-modeled Ze matches the observed Ze. Large differences, however, will likely lead to nontrivial retrieval errors.

For the 14 February anomaly, in addition to the evidence mentioned briefly in

the manuscript, a collocated X-band Doppler radar (McGill University's VertiX) revealed a bright band at around 1 km AGL with Doppler velocities of around 3 m/s below this level. This is a large fallspeed for dry snow aggregates, but these might have been large, wet aggregates. The VertiX was not in operation for the 2 March anomaly. It's probably not possible to rule out that aggregation was involved in either anomaly.

Yes, there does seem to be underestimation for higher snowfall rates in Figure 7 (now Figure 8), balanced by overestimation to some degree as evidenced by events with positive fractional differences in accumulations shown in Table 1. This suggest there might be a benefit for making the particle model a function of the observed reflectivity, and should be investigated further.

We have revised the statement in the Discussion section (DL439) to be more consistent with this clarification.

We have also added brief commentary in the 4th paragraph of section 4 (DL345).

> 2. As a second point, in my understanding, the Z-S retrieval is less dependent
> on the intercept parameter in the millimeter wavelength than in the centimeter
> region where the Rayleigh approximation is applicable (e.g. Rasmussen et al.
> 2003, BukovcÌ\214icÌ\201 et al. 2018). In the abstract, it is stated on line 14, that
> the PSD intercept is less well constrained by the retrieval, and in Appendix
> on line 488, that the measurements better sensitive to log(N0) could benefit
> the retrieval. Could the authors elaborate, whether hese results refer that
> the dependence on N0 is physically no longer similar significant to the
> retrieval.

Yes, log(N0) is still physically significant to the retrieval, since it does influence Ze and snowfall rate; however log(lambda) is more strongly constrained by the reflectivity measurement than is log(N0). That means that the retrieval must employ either "a priori" information or additional measurements to help constrain log(N0).

> And continuing, would then actually simple averaged Z-S relations provide
> equally good retrieval results, in case the particle type and shape are
> correctly modeled? The retrieval results here are compared to the collocated
> measurements of accumulation with a weather sensor. It would be also
> interesting to see, how different are the snowfall rates with this presented
> optimal retrieval method in comparison to these simple Z-S relations presented
> in the literature such as e.g. Kulie and Bennartz, 2009 or Matrosov, 2007.

Regarding simple, averaged Z-S relationships, please see Figure 7 that has been added to the manuscript. The retrieval gives Z-S relationships that are temperature-dependent and that deviate from the plotted linear relationships obtained from published Z-S relationships. It seems likely that temperature-dependent functions could be developed that mimic the Z-S behavior of the retrieval (but not the information-centric diagnostics). Please see also our response to reviewer #1.

> Small comments:

> In the paragraph starting on line 29, the ground-based radar observations are
> described. It is slightly striking that only MRR related literature references
> are stated, even though this paper is describing the retrieval process
> utilized for the W-band. Was there a certain reason for this choice, why e.g.
> other studies presenting W-band observations were left out? Although, some of
> the radars are mentioned on lines 71-73.

No, there wasn't a conscious decision to omit W-band radars. For experiments

like these, W-band radars, because of their primary application as cloud radars, are more often operated for cloud measurement from aircraft and less often for observing snowfall at the ground. We know that GCPEx and ICE-POP in addition to C3VP all deployed ground-based W-band radars. Two Department of Energy funded experiments (StormVEx, at Colorado's Storm Peak Laboratory and BAECC in Finland), also deployed ground-based W-band radars. A number of studies have used W-band radar observations as part of multi-frequency snowfall retrievals, which are not applicable to this work. Development of snowfall retrievals using single-frequency, ground-based radar observations at W-band is not common. Some work has been done using ground-based Ka-band radars for snowfall (e g., Matrosov et al., 2008, JAMC; Cooper et al., 2017)

We have revised the text to include information about experiments with ground-based W-band radars. (DL38)

> Figure 11. Suggestion to use a different color for FD12P to improve readability.

Done, and symbol sizes were increased to improve clarity. (near DL400)

> Line 381 Suggestion to use a different term for shape of the size
> distributions. Although it is clear that the exponential PSD distribution is
> used here and this term describes the effect of lambda in the metric curves in
> Figure 13, it still is close to the widely used shape parameter µ of gamma
> PSD.

We changed this to say simply that "The size distribution plays a significant role..." (DL411)

> Typos:

> Line 307 should be Vaisala FD12P (Vaisala Oyj, 2002) instead of (Viasala Oyj,
> 2002)

"Viasala Oyj" is changed to "Vaisala Oyj". (DL332)

> Line 336 remove the unit

"unit" removed, thanks. (DL366)

> Rasmussen, R., M. Dixon, S. Vasiloff, F. Hage, S. Knight, J. Vivekanandan, and
> M.  Xu, 2003: Snow Nowcasting Using a Real-Time Correlation of Radar
> Reflectivity with Snow Gauge Accumulation. J. Appl. Meteor., 42, 20â\200\22336,
> https://doi.org/10.1175/1520C3

> Bukovcl\214icl\201, P., A. Ryzhkov, D. Zrnicl\201, and G. Zhang, 2018: Polarimetric Radar
> Relations for Quantification of Snow Based on Disdrometer Data. J. Appl.
> Meteor. Climatol., 57, 103â\200\223120, https://doi.org/10.1175/JAMC-D-17-0090.1.

> Kulie, M. S., and R. Bennartz, 2009: Utilizing spaceborne radars to retrieve
> dry snowfall. Journal of Applied Meteorology and Climatology,48 (12),
> 2564â\200\2232580, doi:10.1175/2009JAMC2193.1.

> Matrosov, S. Y., 2007: Modeling backscatter properties of snowfall at
> millimeter wavelengths.619Journal of the Atmospheric Sciences,64 (5),
> 1727â\200\2231736, doi:10.1175/JAS3904.1.

Matrosov, S.Y., M.D. Shupe, and I.V. Djalalova, 2008: Snowfall retrievals using millimeter-wavelength cloud radars. J. Appl. Meteor. Climatol., 46, 769-777. doi:10.1175/2007JAMC1768.1

Cooper S. J., N. B. Wood, and T. S. L'Ecuyer, 2017: A variational technique to estimate snowfall rate from coincident radar, snowflake, and fallspeed observations. Atmos. Meas. Tech., 10, 2557-2571, doi:10.5194/amt-10-2557-2017.

[revised manuscript text omitted]